# Smooth Pseudo-Labeling

## Abstract

Semi-Supervised Learning (SSL) seeks to leverage large amounts of non-annotated data along with the smallest amount possible of annotated data in order to achieve the same level of performance as if all data were annotated. A fruitful method in SSL is Pseudo-Labeling (PL), which, however, suffers from the important drawback that the associated loss function has discontinuities in its derivatives, which cause instabilities in performance when labels are very scarce. In the present work, we address this drawback with the introduction of a Smooth Pseudo-Labeling ($SPL$) loss function. It consists in adding a multiplicative factor in the loss function that smooths out the discontinuities in the derivative due to thresholding. In our experiments, we test our improvements on FixMatch and show that it significantly improves the performance in the regime of scarce labels, without addition of any modules, hyperparameters, or computational overhead. In the more stable regime of abundant labels, performance remains at the same level. Robustness with respect to variation of hyperparameters and training parameters is also significantly improved. Moreover, we introduce a new benchmark, where labeled images are selected randomly from the whole dataset, without imposing representation of each class proportional to its frequency in the dataset. We see that the smooth version of FixMatch does appear to perform better than the original, non-smooth implementation. However, more importantly, we notice that both implementations do not necessarily see their performance improve when labeled images are added, an important issue in the design of SSL algorithms that should be addressed so that Active Learning algorithms become more reliable and explainable.

## 1 Introduction

The massive successes of modern supervised Machine Learning techniques in all standard benchmarks in Computer Vision, Foret et al. (2020); Kabir et al. (2020); Dai et al. (2021); Dosovitskiy et al. (2020), have made possible the relaxation for need of supervision. The field of Semi-Supervised Learning (SSL) thus gained considerable attention, as it promises, under some conditions, to lighten the need for annotated data, condition sine qua non for supervision. In such settings, one generally assumes a large dataset, the greatest part of which is non-annotated, the remaining part having been annotated for the relevant task. In our exposition we will fix the task as Image Classification, so as to avoid having to introduce general and complex notions and notation. Annotations, even for Image Classification, can be costly to obtain, not to mention more difficult problems, in some cases demanding man-hours from highly trained professionals. On the other hand, non-annotated images are orders of magnitude cheaper to obtain. SSL thus seeks to alleviate the cost of training by transforming man-hours to machine-time, while keeping the performance of the model at the same level. The core quest of SSL is, thus, to lower the percentage of data that needs to be annotated in order to keep the same performance as if the whole dataset had been annotated, for a given dataset and task.

SSL approaches rely on variants of Siamese networks, where two branches of neural networks consume different views of the same image, augmented to a different degree. Some kind of consistency is imposed at the output, stemming from the intuition that if the augmentations are not so violent as to totally distort the content of the image, the output of the network should remain invariant. A popular strategy was inaugurated by the introduction of Pseudo-Labeling Lee et al. (2013) consisting in using the model being trained to

infer labels for non-annotated data and using them in conjunction with a small amount of annotated data. Crucially, the score of the dominant class at inference on non-annotated data must exceed a given threshold to be considered as reliable enough and define a pseudo-label. Such a cut-off for admitting pseudo-labels introduces a discontinuity in the derivative of the loss. The fact that the loss function is iteratively defined during training leads to an accumulation of discontinuities and can generate instabilities that can be described in terms of a signal-to-noise ratio problem. The signal in this case comes from supervision, which feeds the network with information independent of the current state of the network. Self-supervision (on unlabelled data), which depends on the current state of the network and contributes to the determination of the next state, is inherently noisy (see also the analysis in Tang et al. (2022)), especially so in the early stages of training, before some level of confirmation bias kicks in Arazo et al. (2020). The fact that instabilities due to the accumulation of discontinuities have not yet been observed widely can be attributed to the strength of the signal needed for performances close to the fully supervised baseline to be obtained in most benchmarks.

FixMatch Sohn et al. (2020) is built upon the Pseudo-labelling principle but obtains the pseudo-label by inference on a weak augmentation of the image and learns it on a strong augmentation. It was the first improvement on Pseudo-Labeling to reach a performance close to the fully supervised baseline on CIFAR-10 with only 40 labels, 4 per class, and thus a very weak supervision signal. As show our experiments, in the best cases, FixMatch can reach $\sim 7\%$ error rate, close to the $4\%$ achieved by the fully supervised baseline. However, it can also go as high as $\sim 15.5\%$, presenting a standard deviation of $\sim 3.00\%$ over 6 independent experiments (cf. table 1 or Sohn et al. (2020)). This was indeed observed in Sohn et al. (2020), but not attributed to any particular factor. We establish that this kind of instability can be attributed to a considerable extent to the discontinuity of the derivative of the loss function minimized in Pseudo-Labeling and, consequently, in FixMatch.

We note that such instabilities are not present in the experiments on CIFAR-100 or ImageNet, as performance remains further from the fully supervised baseline, since the dataset is more difficult, even with considerably stronger supervision. Due to stronger supervision, volatility is considerably lower, since the performance of the network becomes less sensitive in the misclassification of a small number of images. The phenomenon is nevertheless likely to appear in the future, if the methods that bring the performance of SSL and supervised methods closer together still rely on non-continuously differentiable (i.e in $C^0 \setminus C^1$) loss functions. It is also bound to create problems in Active Learning scenarios where labels are added during training. In such cases, the correct response of the model to the strengthening of the signal is not guaranteed, even in the absence of discontinuities in the derivative. In our experiment section we see, however, that, even though smoothing the discontinuities does not solve all issues with PL, it does result in a better overall performance.

Subsequent improvements on FixMatch added modules, resulting in additional computational cost, and/or discontinuities, and/or hyperparameters, or hypotheses on label distribution of non-annotated data Zhang et al. (2021), but none addressed the discontinuity issue.

The fact that the loss minimized by algorithms using Pseudo-Labeling Lee et al. (2013) is $C^0 \setminus C^1$ has, to our best knowledge, not been observed so far in the literature on the subject, despite the latter being quite abundant. Minimization of a function whose derivative presents discontinuities is a serious flaw from the point of view of optimization theory, as it can lead to a sensitive dependence of the point of convergence of the implemented minimization algorithm with respect to the irreducibly stochastic factors of training (random initialization, order and composition of batches, etc.). Non-Differentiable Optimization is an active field of research, relying on specific approaches such as subgradients Lemaréchal (1989). Arguably, failure to observe this important drawback in the method is due to the probabilistic and information-theoretic point of view that is predominant in the field of Machine Learning. We argue that, in view of the cost-less improvement of PL that this observation brings about, as well as the success of the Focal loss function in some settings Lin et al. (2020), the optimization point of view should be given more consideration in the design and the construction of machine learning algorithms. We note that neither the Focal loss, nor the SPL loss introduced in this work have any information theoretical content.

We thus focus in the regime of scarce labels and high performance, where SGD should be expected to be less robust due to the discontinuities in the derivative of the loss. The usual way of comparing methods is by comparison of average performance over a certain number of runs and its standard deviation. However,

in our study we found that this is insufficient, mainly due to the fact that standard deviation is agnostic of the sign.

We argue that when one compares two methods over several runs, it is more rigorous to estimate their difference with an actual statistical test, as is usually done in other scientific fields. The Wilcoxon test was introduced in Wilcoxon (1945) and tests whether the median of an increasing array of numbers is significantly non-zero, positive or negative, depending on the particular version of the test (two-sided, one-sided greater or one-sided less). It produces a confidence level, a p-value which measures the probability of having the given result if methods A and B are equivalent, method A is better than B or vice versa (for the two-sided, one-sided greater or one-sided less tests, resp.).

As we discuss in the related work section, §2, a number of models in the literature on both the balanced and imbalanced class setting use in a crucial way knowledge of the class distribution of the relevant datasets. This is information that is, by the very definition of SSL, not accessible, and use of such information is bound to lead to increased performance, but to the detriment of applicability of the method in real-life conditions, where its assumptions are not verifiable.

This kind of assumption is also hardcoded in the benchmarks on which FixMatch is tested in the original article Sohn et al. (2020), where labeled images are selected so that their distribution agrees with the one of the total dataset. In a real-life scenario, the class distribution of the total dataset is unknown and such sampling is impossible. In order to bridge the gap between experimental and applied settings, we added a benchmark with random sampling of labeled images, and compared our method with FixMatch on it. We refrained from testing other models proposed in the literature, since their application would not be out-of-the-box as their assumptions on class distribution of the (un)labeled dataset were not directly verifiable.

In conclusion, the main contribution of this article is to investigate the consequence of using a continuously differentiable loss in a regime of scarce labels for SSL, leading to "smooth pseudo-labelling" (SPL). We thus obtain an algorithm that consistently outperforms the non-smooth version, and is more stable with respect to hyperparameters and the initialization of the network. At the same time, in the more stable regimes, the performance remains at the same level as the original non-smooth method.

A subsequent contribution is to introduce a novel protocol to compare the methods in such settings. Using the Wilcoxon one-sided test, we found that the proposed smoothing improves the performances of FixMatch ten times more significantly ($p \sim 0.0156$) than FlexMatch does ($p \sim 0.1562$).

Finally, we introduce a benchmark where labeled images are selected at random from the entire dataset, as opposed to enforcing the class distribution in the labeled dataset as it is observed in the entire dataset, despite the fact that the latter is inaccessible by hypothesis. In this setting, we observe an unexpected behavior, both for FixMatch and the smooth version introduced herein. It would be expected, at least by the authors, that expanding the labeled dataset should not result in worsening the performance of the model. However, this was observed for both algorithms, even if to a slightly lesser extent for the smooth version. We thus propose that some kind of monotonicity be built in the construction of SSL algorithms based on Pseudo-Labeling, since a posteriori we realized that the good behavior of algorithms with respect to the expansion of the labeled dataset is in fact not guaranteed by intuition, much less by theory.

## 2 Related work

Two main branches can be distinguished in the field of SSL. The first one, heavier in computational cost, relies on Self-Supervised pretraining, using the whole dataset without annotations in order to learn general, possibly even task-agnostic features. The pretrained model then learns the annotated data in a traditional supervised manner and teaches a simpler model its predictions on both annotated and non-annotated data Caron et al. (2021); Grill et al. (2020); Chen et al. (2020). The second one, considerably cheaper in computational cost, consists in a combination of supervised learning for annotated data coupled with self-supervision for non-annotated data, all at the same time Lee et al. (2013); Sohn et al. (2020); Berthelot et al. (2019b). As should be expected, the first type of strategy performs better in more difficult datasets such as Imagenet, but strategies of the second kind have already been successful in reducing the annotation cost to a minute percentage of simpler datasets such as CIFAR-10 or CIFAR-100, Krizhevsky et al. (2009). Naturally, a

combination of both approaches, i.e. using a strategy of the second type in order to train a pretrained model should be expected to boost performance.

To the best knowledge of the authors, all recent improvements on PL in general and on FixMatch in particular Pham et al. (2021); Lee et al. (2022); Yang et al. (2022); Zhang et al. (2022); Xie et al. (2020); Berthelot et al. (2019a); Kim et al. (2021); Li et al. (2020); Xu et al. (2021), have either preserved the issue of discontinuities in the derivative of the loss function or even worsened it by adding thersholding. In any case, the general tendency is to add modules on top of FixMatch, which results in additional hypermarameters and sometimes discontinuities in the derivative. Moreover, the value of some additional hyperparameters is often changed when passing to more difficult datasets without an a priori, i.e. agnostic of the test-set of the new dataset, justification of the new value. All this comes with computational overhead and/or additional hyperparameter fine-tuning.

FlexMatch, Zhang et al. (2021), did obtain high performance in certain benchmarks but this came at the cost of sneaking in a hypothesis on the distribution of classes in the unlabeled part of the dataset. CPL, the pseudo-labeling strategy used in FlexMatch, lowers the threshold for accepting pseudo-labels for classes that are less represented in pseudo-labels. The weaker representation of a class in the number of pseudo-labels is interpreted by the authors of Zhang et al. (2021) as a difficulty of the model to learn the given class. Even though this can definitely be the case, another possible reason can be that the given class is indeed less represented in the dataset. Knowing which of the two is the actual reason can only be known for sure (as is assumed in the construction of CPL and therefore of FlexMatch) only by annotating the unlabeled dataset. For the heuristic to work, therefore, a hypothesis on class distribution is indispensable, and the one imposed in Zhang et al. (2021) is uniform class distribution. In §3 we argue that FlexMatch is for this reason not an SSL algorithm on par with the rest in the literature, and in §4.3 we establish to what point FlexMatch depends crucially on the alignment of class distribution between labeled and unlabeled datasets. These observations and results question the validity of the remark of the authors of FlexMatch concerning the improvement on the Imagenet 10% benchmark

> This result indicates that when the task is complicated, despite the class imbalance issue (the number of images within each class ranges from 732 to 1300), CPL can still bring improvements.

Firstly, this is as an explicit admission of the fact that class distribution is assumed to be uniform as we were able to find in the article, another one being the discussion in paragraph "Comparison with class balancing objectives". We were unable, however, to find a trully explicit mention of this assumption in the construction of the algorithm, or in the abstract. In the abstract the fact that

> CPL does not introduce additional parameters or computations (forward or backward propagation)

is mentioned, but the fact that additional assumptions are imposed on the (unlabeled) dataset is omitted. We think that making this issue clear is important for the community, as it seems to have gone under the radar of both the peer-review process and the general knowledge of the community. We would expect this kind of assumptions to be made explicit by the authors of the paper whenever they are introduced.

We note that in the CIFAR-100 and Imagenet 10% benchmarks FixMatch (and FlexMatch, as well) collapses a number of classes, i.e. the corresponding columns in the confusion matrix have 0.00 entries, which mechanically implies that some classes are overloaded. Imposing uniformity of class distribution with pseudo-label purity of the order of 60% should be expected to remove more False Positives from the overloaded classes than True Positives, since already random selection is a TP with a rate of only $\sim 0.1\%$. Transferring FP to the formerly collapsed classes will then, in an almost mechanical way, entail some improvement in pseudo-label purity and consequently in accuracy on the test-set, even by pure chance. The reasons for the marginal improvement of the error rate by 2%, from $\sim 43.5\%$ to $\sim 41.5\%$, in the Imagenet 10% benchmark should be sought in an adapted variant of this argument and not assumed to indicate robustness of CPL with respect to class imbalance in any way.

In order to establish the cruciality of this assumption for the improvement of performance of FlexMatch over FixMatch, we needed to address the class imbalance regime in order to establish empirically that FlexMatch overfits the standard benchmarks by hardcoding the uniformity of class distribution in the method. In the setting where datasets are not a priori supposed to have uniform class distribution, as is the case in the CIFAR datasets or Imagenet, recent advances include Kim et al. (2020); Lai et al. (2022); Wei et al. (2021).

In Kim et al. (2020) the class distribution of both labeled and unlabeled data is by construction assumed to be geometric, and a very quick sensitivity analysis is carried out only with respect to the ratio of the geometric progression. In particular, when the ratio of the geometric progression is assumed to be the same for both datasets, the class distribution is assumed to be identical, which is a very strong and unverifiable hypothesis. Moreover, it is assumed that the ordering of classes is the same for the labeled and unlabeled datasets, a hypothesis that cannot be verified, and can only be assumed to be reasonable if the labeled dataset is a large proportion of the total data. The method is validated only with abundant labeled data, and our study of FlexMatch in §4.3 already shows the limits of methods based on such hypotheses when subjected to the stress test of scarce labels.

In Lai et al. (2022), the authors claim that assuming identical ordering of classes with respect to frequency in the labeled and unlabeled datasets entails no loss of generality. This is an obvious mistake, and the ordering can only be trusted (still with a certain probability of error) again when labeled data are assumed abundant. Again, class distributions are assumed to follow the same mono-parametric law (geometric distribution) and the only, limited, sensitivity analysis is carried out with respect to this unique parameter. Again, 1/3 of the dataset is assumed to be labeled, so that the assumptions become plausible, even though no calculation is provided as to the reliability of the hypotheses when labeled data are drawn *iid* from the total dataset.

In Wei et al. (2021) at least 10% of the dataset comes with labels and to our best understanding the class distribution is by construction geometric and the sampling rate for keeping pseudo-labels is also very conveniently supposed to be geometric of opposite monotonicity, while no sensitivity analysis with respect to this hypothesis is included.

We see, therefore, that improvements on FixMatch in the class imbalanced setting either add assumptions considering the class distribution of the unlabeled dataset in the same way as FlexMatch, or they add modules, in which case they need more hyperparameters, or eventually both.

Needless to say, the only way in which exact knowledge of the class distribution can be obtained is by labeling the whole unlabeled dataset, which defeats the purpose of SSL and would beg the question "why not use the labels instead". In the case where the unlabeled dataset is indeed assumed to be unlabeled, as the wording suggests, only an estimation of the class distribution of the unlabeled dataset is available. The quality of the estimation, as can be seen by simple probability arguments, depends crucially on the proportion of labeled images compared with the size of the dataset, and deteriorates as the proportion becomes small. An exhaustive sensitivity analysis with respect to the misalignment between estimation and reality is then needed in order to establish the usefulness of the method in real-life scenarios where the dataset is given and a part of it is selected iid and then annotated, as opposed to the current practice of constructing the labeled and unlabeled datasets the one next to the other assuming some shared properties.

To the end of clarifying the confusion in the order by which the objects in SSL are given (first the whole dataset and then the labeled part ), and to bridge the gap between benchmarks and real-life conditions, we introduce the CIFAR-10_random benchmark, in which we only impose a number of labels and no uniformity of class distribution or any other assumption on the labeled dataset.

Another direction in PL is that of selecting pseudo-labels according to a measure of the confidence of the network on the reliability of the pseudo-label, as e.g. in Kim et al. (2022); Gong et al. (2021). Such methods introduce additional modules and thresholding (with respect to the confidence score), and, as in Gong et al. (2021), in some cases the dependence on the additional hyperparameters seems to be very delicate.

In Huang et al. (2021) the labeled and unlabeled data have different distribution for classes (label shift) and the domain of images (feature shift). The authors thus identify the data shared between labeled and unlabeled sets by estimating domain similarity and label prediction shift between both sets. Thanks to

this adaptation of the domain of unlabeled data on this shared data, the pseudo-labeling is improved in an open-set context.

In Huang et al. (2023), the authors use cross-sharpness regularisation in order to better exploit the unlabeled data. This, however, results in an algorithm of increased complexity, since an estimation of the worst-case model for the empirical risk on labeled data needs to be calculated, and in additional hyperparameters for the model. The authors were unable to find an explicit mention of the number of runs per experiment in the paper, but the low volatility for FixMatch in the CIFAR-10-40 benchmark would point to 3 runs, a protocol that we observed to be insufficient in the current work.

In Chen et al. (2022), the authors decouple the classification head producing the pseudolabels from the one learning them, keeping the same backbone for feature extraction for both. The head producing the pseudo-labels is selected via the optimization of a min-max type loss. However, in the experiment section no volatility is presented. To the best of our understanding, in table 1 of the paper, no standard deviation per dataset is presented which would seem to indicate that the results feature only one run per experiment. In the last column of the table, the average performance over different datasets is calculated (without reporting any standard deviation), which is not a meaningful quantity. If this is indeed the case, the practice is problematic, as incommensurable quantities are averaged without any justification.

In Zhang et al. (2023) a version of consistency regularization is applied to the domain generalization problem. In such a case, the network also needs to decide whether the sample belongs to a known or an unknown class. The training procedure contains a number of introduced discontinuities in the loss, in the choice of the thresholds, but also in the use of discrete quantities, such as the number of samples predicted as known classes which is used as a divisor. It would be interesting to smooth out these discontinuities to see how it influence the performances.

In Tang et al. (2022), the authors average the predictions of a larger number of classifiers in order to obtain the pseudo-labels. To our best understanding, they, too assume uniform label distribution, as a part of the loss minimized brings the empirical label distribution closer to the uniform one. The dependence of the method on the actual number of classifiers is unclear since it drops then rises.

In Chen et al. (2023), the authors also use the negative pseudo-labels on images with a positive pseudo-label. They force a uniform distribution of the negative classes, which, however, introduces an additional dependence of the loss on the current parameter of the network, see §3.3. It would be interesting to apply our smoothing technique on the loss proposed in the reference. However, the Adaptive Negative Learning strategy seems difficult to smooth. Evaluation is on only 3 folds, and a question arises as to whether the choice of uniform distribution for negative classes is a good choice in an unbalanced class distribution situation.

Finally, since our method has the same computational overhead and hyperparameters as FixMatch and no additional assumptions, and since, more crucially, we improve one of the bases upon which FixMatch is constructed (namely PL), we will only compare our method only with FixMatch. Benchmarking the other methods mentioned here above would demand a significant amount of work for lifting or, to the least, adapting their assumptions using only the knowledge provided by the labeled data.

## 3 Method

We will introduce the new loss function in §3.6 after having established some notation and discussed the issue of discontinuity of the derivative in PL Lee et al. (2013).

### 3.1 Notation

Let $\mathcal{D} \subset \mathbb{R}^d$ be a dataset, split in $\mathcal{L}$, the annotated part, and $\mathcal{U}$, the non-annotated one, so that $\mathcal{L} \cap \mathcal{U} = \emptyset$ and $\mathcal{L} \cup \mathcal{U} = \mathcal{D}$. We consider a classification problem with $n$ classes, $n \geq 2$, and $\mathcal{Y} \in \mathcal{L} \times \{0,1\}^n$ the one-hot labels for images in $\mathcal{L}$. The test-set is denoted by $\mathcal{T}$. We also define the function GT assigning to the image $x \in \mathbb{R}^d$ its ground-truth label GT$(x)$. This function is defined on $\mathcal{L} \cup \mathcal{U} \cup \mathcal{T}$ but accessible only on $\mathcal{L}$ for training and on $\mathcal{T}$ for inference. We therefore have that for every $(u,l) \in \mathcal{Y}$, GT$(u) = l$, but for $x \in \mathcal{L}$, GT$(x)$ is inaccessible during training.

Throughout the article, we consider $f_\theta$ to be a family of models with softmax output, parametrized by $\theta \in \mathbb{R}^m$, with a fixed architecture. The goal of SSL is to use $\mathcal{L}$, $\mathcal{Y}$ and $\mathcal{U}$ so as to obtain a value $\theta_0 \in \mathbb{R}^m$ for which the accuracy of $f_{\theta_0}$ on $\mathcal{T}$ is maximized.

Under this definition, FlexMatch Zhang et al. (2021) is not an SSL algorithm as it makes a hypothesis on the values of $\mathrm{GT}(\mathcal{U})$, namely on the distribution of labels

$$\mathbb{E}_{u \in \mathcal{U}}(\mathbb{1}(\arg\max(\mathrm{GT}(u)) = i)), i \in [0, \cdots, n-1]. \tag{1}$$

In the FlexMatch article this distribution is supposed to be uniform and equal to $\frac{1}{n}$, and in our experiment section, §4 we establish that FlexMatch depends crucially on this assumption.

Finally, we will denote by CE the Cross-Entropy loss function, defined by

$$\mathrm{CE}(x, y) = -\sum_{y_i = 1} \log x_i, \tag{2}$$

where $x = (x_i)$ and $y = (y_i)$ are both vectors in $\mathbb{R}^n$. The vector $x$ satisfies $x_i \geq 0$ and $\sum x_i = 1$, and $y$ is a one-hot label vector.

## 3.2 Pseudo-Labeling

Let $\tau > \frac{1}{2}$. PL, as introduced in Lee et al. (2013), ignoring batching and regularization, consists in minimizing

$$L_{PL}(\theta; \theta, \tau, \mathcal{L}, \mathcal{U}) = \mathbb{E}_{x \in \mathcal{L}}[\mathrm{CE}(f_\theta(x), \mathrm{GT}(x))] - \lambda_u \mathbb{E}_{x \in \mathcal{U}}[\mathbb{1}(f_\theta(x) > \tau) \log(f_\theta(x))] \tag{3}$$

with respect to $\theta$ for $x \in D$, where

$$L_{PL}(\theta; \varphi, \tau, \mathcal{L}, \mathcal{U}) = \mathbb{E}_{x \in \mathcal{L}}[\mathrm{CE}(f_\theta(x), \mathrm{GT}(x))] - \lambda_u \mathbb{E}_{x \in \mathcal{U}}[\mathbb{1}(f_\varphi(x) > \tau) \log(f_\theta(x))] \tag{4}$$

and $\varphi$ has been set equal to $\theta$, i.e. the network deciding the pseudo-labels and the one learning them (as well as the Ground Truth labels) share their parameters.

The hyperparameter $\lambda_u > 0$ is the weight of the loss on unlabeled images. The hyperparameter $\tau \in (0.5, 1]$ is the threshold for accepting a pseudo-label for any given image. The first factor is the supervised loss on labeled data, and the second one the self-supervised loss on the unlabeled data.

The intuition behind the method is that, if $\mathcal{L}$ is large enough and if $\tau$ is close enough to 1., then

$$\frac{1}{\#\mathcal{U}} \sum_{u \in \mathcal{U}} \mathbb{1}(\arg\max(f_\theta(u) > \tau) = \arg\max \mathrm{GT}(u)) \approx 1 \tag{5}$$

i.e. the coverage of the unlabeled set by pseudo-labeled images and the purity of pseudo-labels will be high, so that the network will extrapolate sufficiently and correctly from the labeled dataset and will eventually learn correctly the whole dataset.

The derivative of $L_{PL}$ is discontinuous at all images $x$ and parameters $\theta$ such that $\max f_\theta(x) = \tau$, see fig. 1. $L_{PL}$ is, thus, in $C^0 \setminus C^1$.

## 3.3 A remark on the dependence of $L_{PL}$ on $\theta$

The $L_{PL}$ loss function, as well as the rest of the loss functions defined in this paper and, in general, all loss functions based on pseudo-labeling, depend on $\theta$, the parameters of the network, in two ways. The parameters of the network enter the loss function as differentiated variables in

$$\theta \mapsto \mathbb{E}_{x \in \mathcal{L}}[\mathrm{CE}(f_\theta(x), \mathrm{GT}(x))] \tag{6}$$

and in

$$\theta \mapsto \mathbb{E}[\mathbb{1}(f_\varphi(x) > \tau) \log(f_\theta(x))], \tag{7}$$

where $\varphi$ in equation 7 is considered as a non-differentiated parameter equal to $\theta$. This accounts for the first occurrence of $\theta$ in the signature of $L_{PL}(\theta; \theta, \tau, \mathcal{L}, \mathcal{U})$. The parameters $\theta$ also enter the loss as non-differentiated parameters in the mapping

$$\varphi \mapsto \mathbb{E}[\mathbb{1}(f_\varphi(x) > \tau) \log(f_\theta(x))] \tag{8}$$

determining the cut-off of pseudo-labels. This accounts for the second occurrence of $\theta$ in the signature of $L_{PL}$ since $\varphi \equiv \theta$ during training.

The double occurrence has an important consequence, as it results in a long dependence of each gradient descent step with respect to all previous steps. In particular, when discontinuities are present in the loss function, they accumulate all along the iteration of gradient descent, resulting in a very complicated dependence of the final value of the parameters of the network with respect to their values at all intermediate steps.

More precisely, since training a neural network consists in the iterative minimization of a loss function, the loss function $L_{PL}$ at step $i$ depends on the value of $\theta$ at step $i-1$ on the side of non-differentiated parameters: at step $i$, the parameters $\theta_i$ of the network are updated following

$$\begin{aligned} \theta_{i+1} &= \theta_i - \eta \nabla L_{PL}(\theta_i; \theta_i, \tau, \mathcal{L}, \mathcal{U}) \\ &= \theta_i - \eta \nabla L_{PL}(\theta_i; \theta_i), \end{aligned} \tag{9}$$

where we have dropped the dependence on $\tau$, $\mathcal{L}$ and $\mathcal{U}$ as they are fixed during training. However, the same relation holds at step $i$, namely

$$\theta_i = \theta_{i-1} - \eta \nabla L_{PL}(\theta_{i-1}; \theta_{i-1}). \tag{10}$$

Since the second occurrence of $\theta_i$ in equation 9 is not differentiated and the dependence of $L_{PL}$ on this occurrence is non-trivial, we can substitute this second occurrence of $\theta_i$ by its value given by equation 10 and obtain

$$\theta_{i+1} = \theta_i - \eta \nabla L_{PL}(\theta_i; \theta_{i-1} - \eta \nabla L_{PL}(\theta_{i-1}; \theta_{i-1})). \tag{11}$$

This relation is made possible by the second occurrence of the parameter $\theta$ in eq. equation 3, and the iterative nature of gradient descent. This results in the equations equation 9 and equation 10 giving the same value when differentiated, while they depend on different quantities.

One can then continue iterating this substitution backwards once again, by replacing $\theta_{i-1}$ by the corresponding value in the occurrence of $\theta_{i-1}$ itself, and in its second occurrence in $\nabla L_{PL}(\theta_{i-1}; \theta_{i-1})$, resulting in

$$\theta_{i+1} = \theta_i - \eta \nabla L_{PL}\left(\theta_i; \theta_{i-2} - \eta \nabla L_{PL}(\theta_{i-2}; \theta_{i-2}) - \eta \nabla L_{PL}(\theta_{i-1}; \theta_{i-2} - \eta \nabla L_{PL}(\theta_{i-2}; \theta_{i-2}))\right), \tag{12}$$

In this derivation we have ignored regularization and momentum (for the details related to momentum, see §4.3.4). The loss function is therefore defined in an iterative manner, as $\theta_i$ itself is obtained by the same recursive relation, and it's turtles all the way down (fortunately, down only to zero). This fact introduces a very delicate dependence of $\theta_i$ on all previous $\theta_{i-1}, \cdots, \theta_0$ with discontinuities accumulating all the way.

Consequently, any discontinuities in the derivative accumulate as $i$ grows, rendering the algorithm very sensitive to the inherent stochasticity of training (initialization, batch order and composition).

This complication is inherited by all applications of PL known to the authors, and, as will be established in the experiment section, poses serious instability issues in the limit of weak supervision and high performance. Regarding our signal-to-ratio terminology, see §1, we can now formally distinguish between the signal, given by the supervised loss, independent of $\theta$, and the self-supervised loss, depending intractably on $\theta_0$ as the number of iterations $i$ grows. This intractable dependence is inherent in the method, but the accumulation of discontinuities in the derivative and their extremely delicate mitigation by EMA along the way are not.

### 3.4 Smooth Pseudo-Labeling

Let us introduce the function $\Phi : [0, 1] \to [0, 1]$,

$$\Phi(\sigma; \tau) = \begin{cases} \frac{\sigma - \tau}{1 - \tau}, & \tau \leq \sigma \leq 1 \\ 0, & 0 \leq \sigma \leq \tau, \end{cases} \tag{13}$$

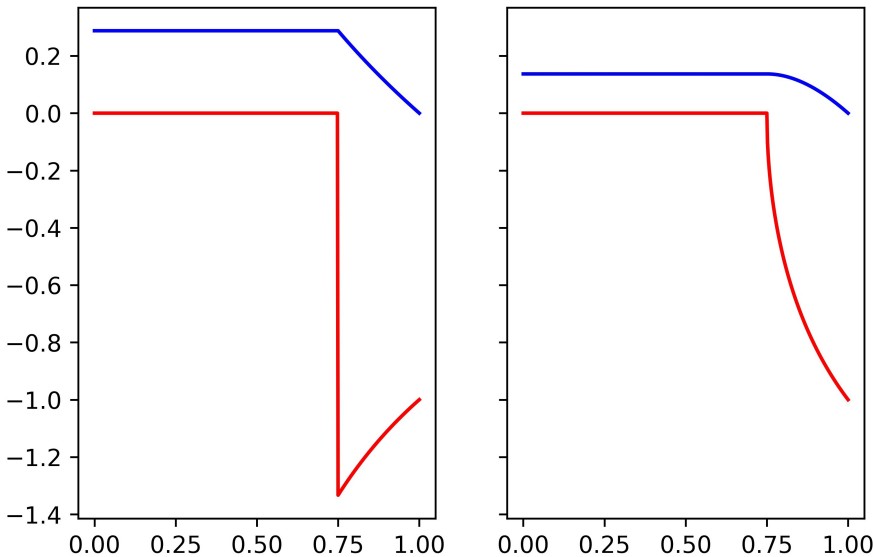

Figure 1: Loss functions minimized by PL (left) and SPL (right) in blue, and their derivatives in red, for $\tau = 0.75$.

or, in more traditional notation, $\Phi(\sigma; \tau) = \mathrm{ReLU}(\frac{\sigma - \tau}{1 - \tau})$. We also define the stop-gradient operator, $\mathrm{sg}(\cdot)$, whose arguments enter calculations as parameters and not as variables differentiated in backpropagation. Hence the function

$$L_{SPL}(\theta; \theta, \tau, \mathcal{L}, \mathcal{U}) = \mathbb{E}_{x \in \mathcal{L}}[\mathrm{CE}(f_\theta(x), \mathrm{GT}(x))] - \lambda_u \mathbb{E}_{x \in \mathcal{U}}[\Phi(\mathrm{sg}(f_\theta(x)); \tau) \log(f_\theta(x))] \tag{14}$$

where

$$L_{SPL}(\theta; \varphi, \tau, \mathcal{L}, \mathcal{U}) = \mathbb{E}_{x \in \mathcal{L}}[\mathrm{CE}(f_\theta(x), \mathrm{GT}(x))] - \lambda_u \mathbb{E}_{x \in \mathcal{U}}[\Phi(\mathrm{sg}(f_\varphi(x)); \tau) \log(f_\theta(x))] \tag{15}$$

has continuous derivative, since the factor of the loss increases continuously from 0 to 1 instead of having a jump at $\tau$. The more general form of the loss function can be written as

$$L_{SPL}(\theta; \Phi, \varphi, \tau, \mathcal{L}, \mathcal{U}) = \mathbb{E}_{x \in \mathcal{L}}[\mathrm{CE}(f_\theta(x), \mathrm{GT}(x))] - \lambda_u \mathbb{E}_{x \in \mathcal{U}}[\Phi(\mathrm{sg}(f_\varphi(x)); \tau) \log(f_\theta(x))] \tag{16}$$

where the function $\Phi$ is a free parameter that is subject to the hypotheses laid out in §3.5, but otherwise free to choose. For the rest of the exposition $\Phi = \mathrm{ReLU}$ will be the working hypothesis for the sake of determinacy.

Taking the derivative of the function $\sigma \mapsto \Phi(\mathrm{sg}(\sigma); \tau) \log(\sigma)$ considering $\mathrm{sg}(\sigma)$ as a parameter and integrating with respect to all instances of $\sigma$ yields

$$\widetilde{SPL}(\sigma; \tau) = \begin{cases} \frac{1 - \tau + \tau \log(\tau)}{1 - \tau}, & 0 \le \sigma \le \tau \\ \frac{1 - \sigma + \tau \log(\sigma)}{1 - \tau}, & \tau \le \sigma \le 1, \end{cases} \tag{17}$$

where the constant of integration was chosen so as to resolve the non-essential discontinuity at $\sigma = \tau$. This is the function plotted in fig. 1(b). Hence, minimizing $L_{SPL}$ in 14 is equivalent (at the $C^1$ level) to minimizing

$$\tilde{L}_{SPL}(\theta; \theta, \tau, \mathcal{L}, \mathcal{U}) = \mathbb{E}_{x \in \mathcal{L}}[\mathrm{CE}(f_\theta(x), \mathrm{GT}(x))] - \lambda_u \mathbb{E}_{x \in \mathcal{U}}[\widetilde{SPL}(f_\theta(x); \tau)] \tag{18}$$

The careful reader might have spotted the need for the introduction of the sg operator in the definition of $L_{PL}$, which should read

$$L_{PL}(\theta; \theta, \tau, \mathcal{L}, \mathcal{U}) = \mathbb{E}_{x \in \mathcal{L}}[\mathrm{CE}(f_\theta(x), \mathrm{GT}(x))] - \lambda_u \mathbb{E}_{x \in \mathcal{U}}[\mathrm{sg}(\mathbb{1}(f_\theta(x) > \tau)) \log(f_\theta(x))] \tag{19}$$

in order to avoid the appearance of a Dirac-delta in the calculation of the derivative of $\mathbb{1}(f_\theta(x) > \tau)$ with respect to $\theta$. It is actually the form of equation 19 that is used in implementations and not the original equation 3, and this seems to have gone unnoticed, along with the double dependence of $L_{PL}$ on $\theta$.

### 3.5 The shape of the factor

The shape of the smoothing factor $\Phi$ need not be linear, but it needs to be continuous and increasing. Smoothness of the resulting loss function imposes that $\Phi(\tau; \tau) = 0$, and normalization that $\Phi(1; \tau) = 1$.

The linear model can be naturally embedded in a one parameter family $\mu \mapsto \Phi^\mu(\cdot; \cdot)$, where $\Phi$ is as in equation 13, $\mu \in [0, \infty]$, and the linear model corresponds to $\mu = 1$. As $\mu \to 0^+$, $L_{SPL}$ degenerates to $L_{PL}$ (though training with $L_{SPL}$ does not degenerate to training with $L_{PL}$ since $L_{SPL}(\mu)$ converges to $L_{PL}$ only in a space of distributions). As $\mu \to \infty$, $L_{SPL}$ degenerates to training on unlabeled images for which the score of the dominant class is already 1. This is *not* equivalent to training only on labeled images, as the derivative of the self-supervised loss function on such unlabeled images is equal to $-1 \neq 0$. Degeneracy is formal also in this limit.

In §4.3.1, we explore three important cases, the linear, $\mu = 1$; the quadratic $\mu = 2$, with an increasing convex shape function; and the square root, $\mu = 1/2$, with an increasing concave function. In any case, the function $\Phi$ represents the confidence of the network on the pseudo-label, and its shape the rate at which the factor of the self-supervised loss should increase as the score of the dominant class increases from $\tau$ to 1: slower than linear for the quadratic function, and faster than linear for the square root.

### 3.6 Smooth FixMatch

FixMatch consists in minimizing the loss function

$$L_{FM}(\theta; \theta, \tau, \mathcal{L}, \mathcal{U}) = \mathbb{E}_{x \in \mathcal{L}}[\text{CE}(f_\theta(x), \text{GT}(x))] - \lambda_u \mathbb{E}_{x \in \mathcal{U}}[\mathbb{1}(f_\theta(x_w) > \tau) \log(f_\theta(x_s))], \tag{20}$$

where $x_w$ is a weak augmentation of the image $x$, and $x_s$ is a strong augmentation of the same image. This loss function presents discontinuities at all images $x$ and parameters $\theta$ such that $\max f_\theta(x_w) = \tau$, just as in PL. Replacing the loss function by

$$L_{SFM}(\theta; \theta, \tau, \mathcal{L}, \mathcal{U}) = \mathbb{E}_{x \in \mathcal{L}}[\text{CE}(f_\theta(x), \text{GT}(x))] - \lambda_u \mathbb{E}_{x \in \mathcal{U}}[\Phi(\text{sg}(f_\theta(x_w)); \tau) \log(f_\theta(x_s))] \tag{21}$$

yields a smooth loss function, again just as in PL. As before, we consider the auxiliary function

$$L_{SFM}(\theta; \varphi, \tau, \mathcal{L}, \mathcal{U}) = \mathbb{E}_{x \in \mathcal{L}}[\text{CE}(f_\theta(x), \text{GT}(x))] - \lambda_u \mathbb{E}_{x \in \mathcal{U}}[\Phi(\text{sg}(f_\varphi(x_w)); \tau) \log(f_\theta(x_s))] \tag{22}$$

and identify $\varphi$ with $\theta$. The loss function depends on two parameters, making visualization less attractive, but the principle is the same as in PL. The function $\Phi$ can, as in the pure PL loss, be considered as a free parameter, and its influence will be explored experimentally in §4.3.1.

In the experiment section we compare Smooth FixMatch with the original implementation and establish that in the limit of weak supervision and high performance the smooth version performs significantly better, while the same level of performance is maintained when supervision is stronger and/or performance is far from the fully supervised baseline. Concerning the shape of the factor $\Phi$, we found that the linear function performs better overall, when compared with FixMatch with the same set of hyperparameters. Different hyperparameters gave good results for the quadratic and the square root factors, so their usefullness should not be excluded. Naturally, the quadratic factor performed better when the threshold was lowered, and the square root performed better with a higher threshold, see table 1 and fig. 2.

In FixMatch, the weight $\lambda_u$ was set equal to 1 for all experiments. In order to keep results comparable, we kept the magnitude of the equilibrium value of the unsupervised loss of $L_{SFM}$, $\ell_{SFM}$, equal to the one of $L_{FM}$, $\ell_{FM}$, see §C.1 for the relevant calculation.

### 3.7 Learning the smoothness factor

It is tempting to add the factor

$$L_\Phi = -\lambda_\Phi \Phi(\text{sg}(f_\theta(x_w)); \tau) \Phi(f_\theta(x_w); \tau) \tag{23}$$

to the $L_{SFM}$ loss function, so that the factor can be maximized, as it learns $x_w$, contrary to $x_s$ learned in $L_{SFM}$. The smoothness of $L_\Phi$ is once again assured by the multiplication factor with sg.

FixMatch established that the network benefits more from learning the strong augmentation of the image than the weak one. In §C.2 of the supplementary material we present a calculation producing the value $\bar{\lambda}_\Phi$ beyond which the model is more incited to learn the weak augmentation than the strong one. A natural choice for $\lambda_\Phi$ is thus in the interval $[0, \bar{\lambda}_\Phi]$.

Our related ablation study (cf. §A.3.7) showed instabilities when the upper limit is reached. Surprisingly, even values of the order of 1% of $\bar{\lambda}_\Phi$ were detrimental to performance. We thus kept the default value $\lambda_\Phi = 0$ and did not add this loss to Smooth FixMatch.

## 4  Experiments

In the experiment section, we use FixMatch as baseline, a common practice in current literature in the subject, in order to test the gain of imposing the continuity of the derivative of the loss function without the additional complications of added modules and hyperparameters, which, to boot, worsen the problem of the discontinuity of the derivative of the loss.

We thus compare our method with the baseline, but also with FlexMatch, holding the SOTA performance in the benchmark CIFAR-10-40 which is of particular interest in our study, due to the high volatility of results. We establish the superiority of our method, and observe that FlexMatch presents the same kind of volatility as FixMatch in the standard benchmark, but, what is more important, is extremely sensitive with respect to the statistics of class distribution, contrary to our method and to the original application of FixMatch.

In all tables, we report the error rate of the last checkpoint. Sohn et al. (2020) reports the median over the last 20 checkpoints, while Zhang et al. (2021) reports best error rate, both not accessible in real conditions. In particular, FixMatch presented some sudden and significant drops in a few experiments, see §4.4, establishing that the best performance is not a good measure (and reminding why it is not used in the literature).

All tables with detailed results, as well as some additional ablation studies, are presented in the appendix §A, in the same order as in §4 of the article, for ease of cross-reference.

Our runs of FixMatch and FlexMatch use and adaptation of the codebase of Zhang et al. (2021). Apart from adding our implementation of Smooth FixMatch, most notably we ensured that batch composition is correctly controlled by the random seed, and we separated the random condition for the creation of the labeled datasets of the different folds from the one controlling training (initialization of the backbone and batch composition) in order to bring the evaluation protocol closer to real-life conditions, where the constitution of the labeled dataset is independent of training.

### 4.1  CIFAR-10-40

This benchmark consists in keeping 4 labeled images from each of the 10 classes of the dataset CIFAR-10, 40 in total. In order to check the compatibility of our results with both the original FixMatch paper, Sohn et al. (2020), and the FlexMatch paper, Zhang et al. (2021), we run 6 experiments in total, indexed $0-6$: $0-2$ for FlexMatch and $1-5$ for FixMatch. Our results for indexes $0-2$ agree with those of Zhang et al. (2021), and those indexed $1-5$ agree with those of Sohn et al. (2020). However, the additional experiments indexed $3-5$ give significantly worse results for FlexMatch than in Zhang et al. (2021), see tables 1 and 7. We note that, while the results for FixMatch and Smooth FixMatch were reproducible after the necessary modifications assuring control of batch composition by the random condition, we were unable to get reproducible results for FlexMatch.

For FixMatch we observe two basins of attraction for the accuracy, in accordance with Sohn et al. (2020): one low, $\sim 83-87\%$ (roughly 1 in 6 experiments, accounting for the lower mean and high std of FixMatch and for the very high std of FlexMatch) and one high, $> 90\%$. For FixMatch, convergence to either basin seems to be determined by both the labeled dataset as well as the random condition, as observed in the original paper.

|  | mean $\pm$ std | gain wrt FixMatch | p-value |
|---|---|---|---|
| FixMatch* | 9.28 $\pm$2.95 | | |
| Smooth FixMatch | **7.24** $\pm$**1.94** | **2.04** $\pm$1.27 | **0.015625** |
| Smooth FixMatch-sq[†] | 7.71 $\pm$2.98 | 1.57 $\pm$**0.91** | 0.031250 |
| Smooth FixMatch-sqrt[‡] | 7.72 $\pm$2.60 | 1.55 $\pm$1.02 | **0.015625** |

Table 1: Error rate on 6 folds of CIFAR-10 with 40 labels of our runs of FixMatch without and with the smoothness factor. The p-value line corresponds to the result of the Wilcoxon one-sided test for the given method having lower error rate than FixMatch, which is used as baseline (median being significantly positive, lower is better). * Our runs. [†] Factor shape is quadratic, threshold set at 0.82. [‡] Factor shape is square root, threshold set at 0.98.

Convergence to the low basin of attraction occurs when one out of the 10 classes of the dataset is totally or partially collapsed, i.e. when the corresponding column in the confusion matrix has 0 or small ($\lesssim 0.20$) elements. This can happen either because of bad representativeness of labeled examples, unfavorable batch order and/or composition, unfavorable initialization or a combination of the above. The instabilities observed in FixMatch training are a manifestation of a delicate competition between the tendency of the model to be more expressive and not have null columns in its confusion matrix, and the $L^2$ regularization factor, which can overcome the tendency for expressiveness and totally or partially collapse one class when the drive for expressiveness, coming from supervision or CR, is overcome. The phenomenon of collapsed classes is stronger in the more complicated datasets (CIFAR-100 and Imagenet) where the confusion matrices are very sparse, which accounts to a considerable extent for the high error rates.

In practice, the collapse can be detected and ad-hoc solutions could be given such as pausing training and extending the labeled set, but this could also be problematic in the present state of the art, see our discussion in §4.5. A better understanding of the behavior of SSL algorithms seems to be necessary so that Active Learning can be based on more solid foundations.

Continuity with linear coefficient improves performance in the high basin of attraction and approaches the low basin to the high one. Continuity with square factor and low threshold improves performance in both basins, but the low basin remains in the same window. Likewise for square root and higher threshold. Shapes alternative to linear (square and square root) with the standard threshold switched experiments converging to the high basin for FixMatch to the low basin for Smooth FixMatch with the respective factor shape.

Some experiments kept learning until the very end of training, hinting that a higher learning rate or a different scheduler with less steep decline of the learning rate could be more adapted. However, since the fact that our method improves on FixMatch at a significant level is well-established without this eventual additional improvement, we favoured more important ablation studies instead.

## 4.2  Strong supervision regime

We verify that in the more stable benchmark of CIFAR-100 with 2500 labels, 25 per class, as well as in the Imagenet-10% benchmark ($100K$ labeled out of $\sim 1000K$ images) our improvement does not deteriorate performance when applied on FixMatch. The performance of PL-based methods in these benchmarks is far from the fully-supervised baseline, and supervision is considerably stronger. This combination results in a std $\sim 0.2\%$ for FixMatch on CIFAR-100-2500 and $\sim 0.5\%$ on Imagenet, an order of magnitude less than the one observed in CIFAR-10-40 experiments. The error rates are $\sim 30\%$ and $\sim 45\%$, respectively, quite

far from full supervision. The results are in tab. 4.2 for CIFAR-100, with detailed results in §A.2 of the appendix, and in §B.2 of the appendix for ImageNet.

| | C100-2500 | | | | | |
|---|---|---|---|---|---|---|
| | FixMatch*† | FlexMatch*† | Smooth FixMatch† | FixMatch*†† | FlexMatch*†† | Smooth FixMatch†† |
| mean ± std | 30.35 ±0.19 | 30.17 ±0.26 | 30.11 ±0.48 | 29.14 ±0.21 | 28.27 ±0.44 | 29.21 ±0.28 |
| gain wrt FixMatch | | 0.18 ±0.28 | 0.24 ±0.30 | | 0.87 ±0.59 | −0.07 ±0.10 |
| p-value | | 0.25 | 0.25 | | 0.125 | 0.8413‡ |

Table 2: Accuracy on 3 folds reporting error rate of last checkpoint. * Our runs. † Weight decay rate 0.0005 as for CIFAR-10. †† Weight decay rate 0.001 as in experiments carried out in Sohn et al. (2020); Zhang et al. (2021). ‡ Two out of three are ties, p-value not reliable.

The instabilities caused by the discontinuity of the derivative are thus not significant in this regime, and we only need establish that our method does not harm performance outside the regime where it is supposed to improve it. The relevant results are presented in the appendix for completeness.

### 4.3 Ablation studies

### 4.3.1 Shape of continuity factor

We put to test shapes for the continuity factor alternative to the simplest linear one, two natural choices being the square root and the square of the linear continuity factor.

The square root attributes higher weights for scores closer to the threshold as it increases more steeply. Experiments with the standard configuration of FixMatch gave a high error rate on seed 5 of CIFAR-10-40, $\sim 14.00$ against 8.01 for FixMatch, while on other seeds it improved on FixMatch. Increasing the threshold to 0.98 from 0.95 was found to consistently improve on FixMatch, see table 1, confirming our intuition. The threshold for more realistic datasets like Imagenet is lower, 0.70 by default for FixMatch, which relaxes the sensitivity of this configuration with respect to the value of the threshold, even though it remains more sensitive than the linear factor.

The square of the linear factor increases slower than the linear one, and thus attributes smaller weights to the loss when the score is close to the threshold. This shape performed worse than FixMatch on seed 2 with standard threshold, giving an error rate of 12.04% against 7.48%. However, lowering the threshold to 0.82 gave the same significance level for improving on FixMatch as the linear factor with the standard threshold, only with a smaller run-wise gain, see table 1.

Obtaining a better performance when lowering the threshold for the square and when increasing it for the square root is easily found to be intuitive.

### 4.3.2 Threshold

We vary the threshold for accepting pseudo-labels on the first fold of CIFAR-10-40, and report results in figure 2. FixMatch is found to depend in a much more sensitive way with respect to the value of the threshold $\tau$ for accepting a pseudo-label than the smooth version with linear factor. The quadratic factor gives better results for lower values of $\tau$, and the square-root better for higher ones, confirming intuition. We provide an indicative plot of linear and quadratic factors in the appendix.

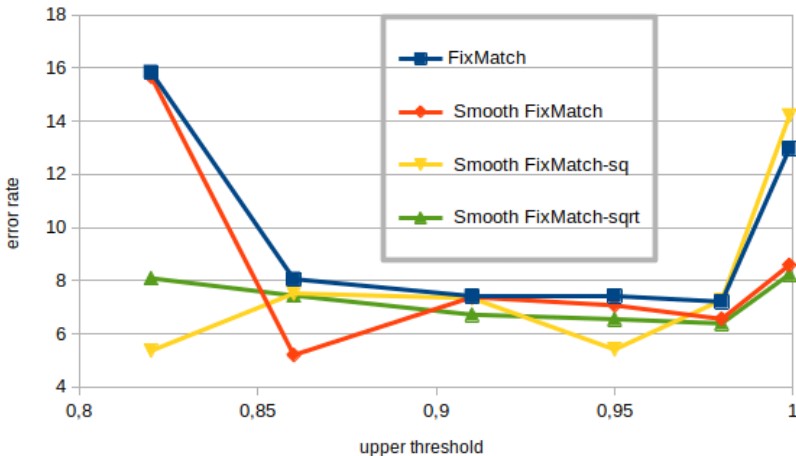

Figure 2: Error rate on the 1st fold of CIFAR-10-40 when the threshold varies. The smooth versions of FixMatch with all three shape factors (linear, quadratic, square root, reliably outperform the baseline.

|  | FixMatch | FlexMatch | Smooth FixMatch |
|---|---|---|---|
|  | 9.30 ±3.43 | 11.59 ±5.90 | **7.78** **±2.71** |
| gain wrt FixMatch |  | −2.28 ±3.05 | **1.51** **±1.94** |
| p-value |  | 0.953125 | **0.109375** |
| drop due to imbalance | **0.03** **±1.83** | 4.80 ±7.15 | 0.55 **±0.89** |
| p-value | **0.421875** | 0.109375 | 0.15625 |

Table 3:     Ablation study on class imbalance. Error rates with one class reduced to 60%. FlexMatch is very sensitive on the class distribution in the unlabelled dataset, while FixMatch and SmoothFixMatch are significantly more robust. Drop due to imbalance is the difference with respect to the performance of the same model without class imbalance (smaller is better). The p-value of drop due to imbalance is the significance level at which the method is affected by the introduction of imbalance (bigger is better).

### 4.3.3   Class imbalance

We introduce a non-alignment between the class distribution of the labeled and unlabeled datasets. For each fold, one class is chosen randomly and its unlabeled dataset is reduced to 60% of the rest of the classes by dropping randomly 40% of the unlabeled images in that class. This is done in order to establish the sensitivity of FlexMatch with respect to the uniformity of class distribution, which shows clearly in our results, reported in table 3.

We see that even in this mild scenario FlexMatch is found to perform worse than FixMatch at a $< 0.05$ significance level, while Smooth FixMatch is found to perform better at a $\sim 0.11$ significance level. FixMatch and Smooth FixMatch do not see their performance drop as much as FlexMatch. This ablation confirms our claim that FlexMatch is not an SSL algorithm on the same standing as others, and, indeed is not an SSL algorithm according to our definition.

### 4.3.4 SGD momentum parameter

In FixMatch, a rapid degradation of performance was observed in their experiments on CIFAR-10 with 250 labels (and therefore considerably stronger supervision signal) when the momentum parameter of the SGD optimizer is increased beyond the default value $\beta = 0.90$.

Momentum is in general beneficial to the stability of convergence because it introduces an exponential moving average (EMA) smoothing of the gradients and reduces noise in the direction of gradient descent. Since the derivative of the loss of FixMatch is discontinuous and, as we have observed, the loss is actually defined in an iterative manner due to its dependence on the state of the model at the each step, momentum in this case results in an accumulation of discontinuities, which becomes more significant as the value of $\beta$ increases. On the other hand, these discontinuities are smoothed out to a certain extent by EMA.

Imposing smoothness on the derivative of the loss function removes the additional stochasticity due to discontinuities, but does not alter the fact that the loss is defined in an iterative manner, and thus inherently highly stochastic.

The trade-off between accumulation of discontinuities, which adds stochasticity in the gradient at a given step, and the benefits of EMA smoothing is far too complicated to analyze in a realistic setting. It should, nonetheless, be expected to introduce a very sensitive dependence on $\beta$ in the limit of weak supervision and high performance when the loss is not $C^1$-smooth.

It is not surprising to observe that increasing $\beta$ beyond a certain tipping point becomes detrimental to performance, and, to boot, in a quite catastrophic manner. This is observed for both algorithms, but seems to harm ever so slightly more the smooth version, which suffers significantly less from randomness in the value and direction of the gradient. This tipping point manifests itself at the value where the benefits of smoothing the noise in the direction of gradient descent via EMA are overpowered by the accumulation of discontinuities (if present) and the long memory of the gradient from iterations where pseudo-labels were less abundant and/or reliable than at the given step. Naturally, the value of the tipping point depends on the strength of supervision signal and the particularities of the dataset in a convoluted way, making it impossible to determine theoretically, but only accessible empirically.

We verify this trade-off hypothesis by repeating the experiment of Sohn et al. (2020) in the more difficult setting of CIFAR-10-40 (instead of 250), comparing the behavior of FixMatch without and with the smoothness factor, and report results in figure 3. The default parameter for the rest of our experiments is $\beta = 0.90$.

What happens when momentum is reduced, adding stochasticity to gradient descent is more interesting. Already for $\beta = 0.85$, instead of the default value $\beta = 0.90$, FixMatch presents instabilities with the model spiking to an error rate $\sim 25.00\%$ once at $\sim 40\%$ of iterations, but then regressing to $\sim 30.00\%$ and never recovering, practically doubling its error rate for a $\sim 5\%$ decrease in $\beta$. The smooth version also saw its performance drop, but by $\sim 30\%$, as opposed to $\sim 90\%$ for FixMatch.

This confirms experimentally the delicate nature of the stochasticity of the loss and its derivative both in the the non-smooth and the smooth setting. At the same time it proves beyond any doubt that non-$C^1$-smooth loss functions are highly problematic in the limit of weak supervision and high performance.

### 4.3.5 Ablation study on initial condition

In Sohn et al. (2020) the authors observed that the model has the same variance across folds, as on the same fold of CIFAR-10-40 with varying random seed. They obtained a mean error $9.03 \pm 2.92\%$ over 5 runs.

We repeated the ablation using the 4th fold, which was the only one to converge to the low basin of attraction for FixMatch in our standard experiments with an error rate of $15.60\%$, cf. table 4.

In our ablation, FixMatch was found to have an error rate of $16.95 \pm 3.27$, while Smooth FixMatch $12.97 \pm 0.78$, a striking difference. The performance of FixMatch in the low basin of attraction seems to be very sensitive when an unfavorable network initialization and/or unfavorable order and composition of batches is coupled with a less representative labeled dataset. We note that the initial condition controls not only the initialization of the network, but also the augmentations applied to the images throughout training. More

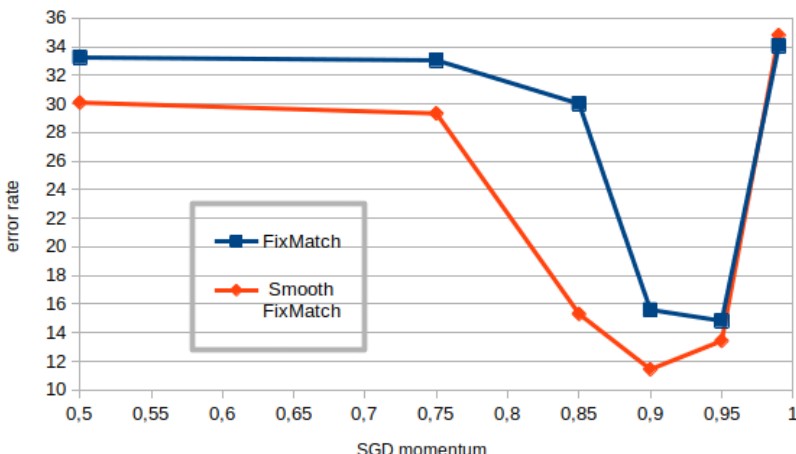

Figure 3: Error rate on the 4th fold of CIFAR-10-40 when SDG momentum varies. The window allowing a good performance for FixMatch is very narrow, but becomes significantly wider upon introduction of the smooth loss function.

precisely, FixMatch lost up to $\sim 7.\%$, giving an error rate as high as $\sim 22.50\%$, while the smooth version of the algorithm did not lose more than $2.\%$. The worst performance of the smooth version is on par with the best one of the original, non-smooth, implementation. This means that the maximum experimentally observed error rate for Smooth FixMatch is as high as the lowest error observed for FixMatch, assuming convergence to the low basin of attraction for the standard CIFAR-10-40 benchmark.

Cross comparison of these results with those of table 7 shows that FixMatch indeed has a comparable statistical behavior across folds and across runs. On the contrary, Smooth FixMatch is less sensitive to the random condition than to the representativity of the labeled dataset, which is a more desirable behavior. The representativity of the labeled dataset, coupled with an unfavorable random initial condition can severely penalize FixMatch, while Smooth FixMatch behaves in a significantly more stable manner, as it does also when the labeled dataset varies. Once again, this proves our main argument, as smoothing out the discontinuities in the derivative of the loss accounts for a considerable reduction of the volatility of the error rate, and a pointwise improvement, when each possible factor introducing stochasticity is taken into account.

|  | mean $\pm$ std |
| --- | --- |
| FixMatch | $16.95 \pm 3.27$ |
| Smooth FixMatch | $\mathbf{12.97 \pm 0.78}$ |

Table 4: Ablation study on random seed: error rate on the 4th fold of CIFAR-10-40 when the random seed varies. The default random seed for our experiments is 2046.

The seed 1917 produced an instability for FixMatch, with a minimum error rate $\sim 10.\%$ reached at $\sim 800K$ iterations, with a final error rate $\sim 17.\%$. Likewise, the best error rate for seed 2001 was $\sim 13.5\%$, reached at $\sim 95\%$ of iterations, but then rose to $\sim 22.5\%$.

Smooth FixMatch with linear factor did not exhibit this kind of instability in the numerous experiments carried out under the same conditions.

## 4.4 Aggregate comparison and some comments

When all our experiments on CIFAR-10-40 on FixMatch versus Smooth FixMatch with linear continuity factor performed under the same conditions are compared run for run, including ablations, Smooth FixMatch gains 2.61 on average on FixMatch, with a standard deviation of 3.28. The p-value for Smooth FixMatch performing better than FixMatch is $4.35e{-}5$. The maximum gain is 14.70, the minimum is $-1.30$. Smooth

FixMatch obtained a better performance in 23 out of 26 experiments, including for reasons of transparency experiments where both performed poorly. These results corroborate the observation that calculation of mean and standard deviation of the run-wise difference in scores is an insufficient measure for the comparison of algorithms, since standard deviation is sign agnostic, which can produce misleading figures when the differences are almost exclusively one-sided.

The experiment with FixMatch on CIFAR-10-40 with the threshold set at 0.82, see fig. 2, had a best error rate of 6.46 attained at $\sim 90\%$ of iterations, but presented an instability and dropped to 15.85%. Similarly, on the same dataset with SGD momentum $\beta = 0.85$, FixMatch spiked to an error rate $\sim 25.00\%$ once at $\sim 40\%$ of iterations, but then regressed to $\sim 30.00\%$ and never recovered. The $\sim 30.00\%$ error rate corresponds to 3 collapsed classes with one column summing up to $\sim 0.37\%$, another up to $\sim 0.20\%$ and a third up to practically 0.00.

Such instabilities were observed only once in our experiments where the continuity factor has been integrated into the unsupervised loss, and the discrepancy between the best error rate and the one of the last checkpoint were insignificant. The only exception was a discrepancy of $\sim 1.5\%$ between best and final error rates for the square root factor and $\tau = 0.82$, which again is in agreement with our intuition. In a considerable portion of our experiments, actually, best error rate was achieved late in training, hinting that the learning rate and scheduling are not optimal for our method, but investigating this direction as well would lead us astray from the main argument of the paper.

In the standard CIFAR-10-40 benchmark, FlexMatch is found to gain 2.10 on FixMatch on average, run for run, a greater average improvement than our 2.04. The standard deviation of FlexMatch's gains on FixMatch is $\sim 5.7\%$ while ours $\sim 1.3\%$. This however still hides the fact that our method is better than FixMatch in every single experiment of table 1, while FlexMatch only in 5 out of 6 (cf. table 7). Such intricacies made necessary the introduction of the Wilcoxon test, which is a more adapted measure of comparison.

### 4.5 Random choice of labeled dataset

### 4.5.1 Presentation of the benchmark

In order to contrast with current practices of hardcoding properties of the benchmark datasets into the learning procedure of proposed methods, as we discussed in our Related Work section, §2, we introduce the following benchmarks which bring evaluation protocols closer to real-world practice.

We consider a dataset, say CIFAR-10, and fix the number of labeled images, say 40. Contrary to the widely studied benchmark CIFAR-10-40, we *do not* chose 4 labels for each class, imitating the uniform class distribution of CIFAR-10, but we randomly choose 40 images to label, uniformly from the whole dataset, and fix 6 such folds, numbered $0 - 5$. This brings the evaluation protocol closer to real-life scenarios, as in practice class distribution is unknown and can only be extrapolated from that of the labeled dataset, subject to an uncertainty that grows as the number of labels becomes smaller.

Indicatively, the class distribution of the 0-th fold features 1 class with frequency 2.5%, 3 classes with frequency 5%, 2 classes with frequency 10%, 3 classes with frequency 12.5% and 1 class with frequency 25%, which is considerably far from the real uniform distribution of 10% for each class. Such phenomena should not be expected to be rare in the regime of weak supervision, provided that the sampling of labeled data is truly random and has not artificial similarities with the whole dataset, imposed by construction.

We note that the 3rd fold has two classes with no labeled image associated to them. Concretely, we think that since the probability of the existence of a class with 0 frequency in the labeled dataset is not negligible, the benchmark with 40 images labeled out of $50K$ is not relevant. We decided to keep the fold in the benchmark nonetheless and train a 10-class classifier on it in order to showcase the difficulties of the regime, technical, conceptual and protocol-related, and for reasons of comparison with the standard benchmark with uniform class distribution imposed on the labeled dataset.

A single fold with one class with 0 labeled representatives persists in the 50 label setting, but all classes are represented in all folds starting from the 60 label setting. Class distribution remains quite far from the uniform 10% for all classes, however, as can be seen in fig 4. For each fold of for each benchmark of 40 up to

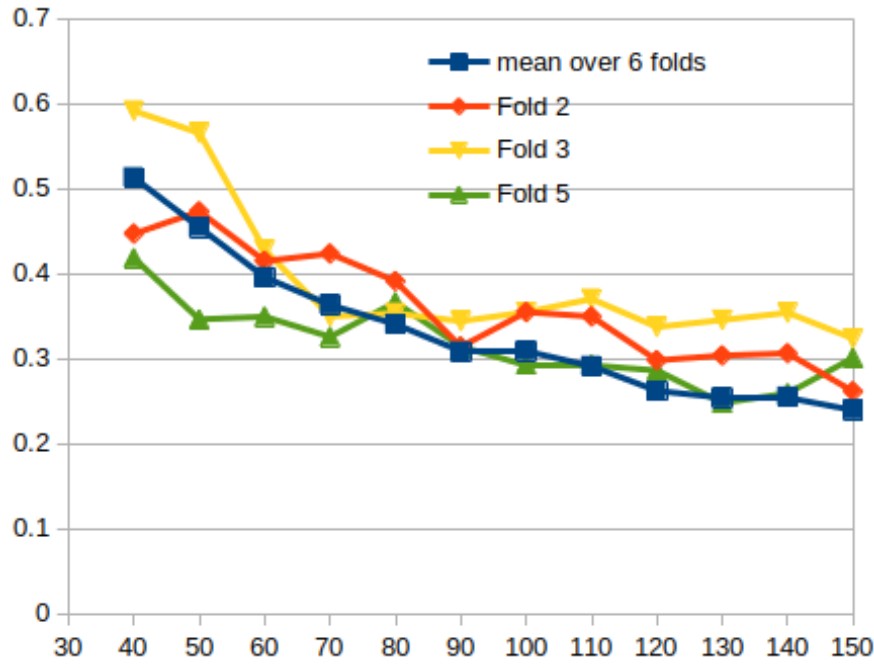

Figure 4: Standard deviation of class frequency from uniform class distribution of CIFAR-10 with random sampling normalized by the uniform frequency 1/10, over 6 folds. The graph shows that with truly random sampling the class distribution of the labeled dataset remains quite far from the true (uniform) class distribution of the whole dataset. We also include the evolution of the standard deviation of 3 folds that proved difficult for the model to converge. In particular, the difficulty of fold 5 shows that standard deviation is a crude measure, and that representativeness of images does play a role in convergence.

150 labels, we calculate the standard deviation of class frequencies around the mean frequency 1/10, and we normalize by 1/10, expressing the standard devation as a fraction of the mean value. The plot features the mean of this normalized standard deviation over the 6 folds of each benchmark as well as that of 3 folds for which the models had difficulty converging. It starts at ∼ 50% of the mean value for 40 labels and decreases slowly to ∼ 25% for 140 labels.

Methods using assumptions on class distribution tested on this benchmark, which can be directly carried over to the imbalanced class setting, can only use the observed class distribution of each labeled fold, and not the uniformity of overall class distribution in our case (or the actual class distribution in the general case).

Extrapolation of the class distribution from a few labeled data is easily seen to be completely unjustifiable in an empirical way. A rigorous calculation of confidence intervals of class distribution should have made part of any method based on such an extrapolation, along with a rigorous sensitivity analysis of the effect of misestimation.

The goal of the benchmark is to reach stable performance close to the fully supervised baseline with the minimal number of labels using only the properties of the each labeled dataset for each run. This protocol imitates an Active Learning scenario for which each time labels are randomly selected and added to the labeled dataset the model is trained from scratch.

The naive expectation for SSL models would be that performance should not worsen when labels are added in this incremental way. However, we observe that CR based on PL behaves badly in this scenario. Smoothing out the discontinuities in the derivative does improve the behavior of the problem, since the smooth model obtains a stable performance close to the fully supervized baseline before the non-smooth one. Smoothness

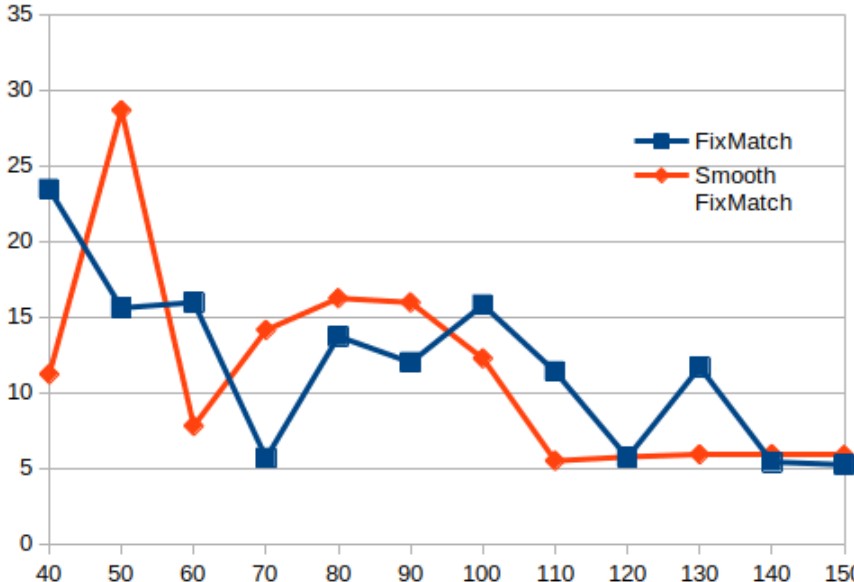

Figure 5: Evolution of the error rate on the 5th fold of CIFAR-10 with random sampling. Both models can regress to higher error rates when the number of labeled images is increased, but this occurs for greater number of labels for FixMatch than for Smooth FixMatch.

of the loss function nevertheless is not the root of the problem, as the smooth version still presents drops in performance when labels are added.

This observation raises the question as to how the model will react if labels are added and training is resumed. Will adding labeled data result in the model correcting its confirmation bias or will it ignore the new labels and stick to its predictions? If our benchmark allows for a prediction, it should be that the smooth version would be faster to correct its biases than the non-smooth one, but both would react in a non-monotonic way.

### 4.5.2   Results

We test Smooth FixMatch against the baseline FixMatch with a progressively increasing number of labels by 10 at each step until one algorithm reaches a stable performance close to the supervised baseline across the 6 folds. The goal is to reach stable performance equal to the fully supervised baseline with the minimal number of labels. Results are presented in table 5.

In the regime where the performance of both models is far from the fully supervised baseline, the performances of the models seem to be comparable. In addition, both models may regress to worse performance upon increase of labeled images. However, Smooth FixMatch reaches the point of diminishing returns earlier than FixMatch. The smooth model obtains a stable mean error rate under 8.00% and a standard deviation smaller than 1% with 120 labels and stays in that regime up to 150 labels. However, FixMatch can regress to a worse performance even in the passage from 120 to 130 labels, and, then reach a competitive error rate at 140 labels, which is better than that of the smooth version for the same number of labels (but worse than that of Smooth FixMatch with 130 labels).

As show the results of table 5 and, in more detail, those of the Excel sheet accessible here, the response to the expansion of the labeled dataset of both FixMatch and Smooth FixMatch, even if to a slightly lesser extent for the latter, is not the expected one. The p-value for the algorithm gaining in performance after labeling $10 \times n$ images (cf. the Excel sheet accessible here for more detailed results) is not consistently low. We indicatively present in fig. 5 the evolution of the error rate on the 5th fold of the benchmark, the one that proved to be the most difficult for the models to solve. We remind the reader that the benchmarks with 40 and 50 labels are in fact irrelevant (as some classes are not represented in the labeled set) but are included only for comparison with the standard benchmark.

| labels | | CIFAR-10 with random sampling | | | | | | | | | | |
| | 40 | 50 | 60 | 70 | 80 | 90 | 100 | 110 | 120 | 130 | 140 | 150 |
|---|---|---|---|---|---|---|---|---|---|---|---|---|
| FixMatch* | 16.34 ±8.23 | 14.83 ±9.23 | 11.54 ±5.02 | 8.02 ±3.71 | 10.15 ±4.73 | 9.26 ±3.93 | 9.56 ±4.27 | 7.49 ±2.94 | 6.01 ±0.70 | 6.69 ±2.26 | 5.64 ±0.64 | 5.63 ±0.43 |
| Smooth FixMatch | 17.51 ±7.21 | 15.37 ±10.06 | 9.09 ±3.99 | 7.67 ±3.21 | 8.08 ±3.76 | 8.91 ±4.84 | 8.40 ±4.03 | 8.35 ±4.02 | 5.96 ±0.78 | 5.35 ±0.26 | 6.12 ±0.88 | 5.61 ±0.48 |
| gain wrt FixMatch | −1.17 ±8.07 | −0.54 ±6.28 | 2.46 ±3.47 | 0.36 ±5.68 | 2.07 ±4.23 | 0.36 ±3.00 | 1.14 ±3.14 | −0.85 ±4.59 | 0.05 ±0.20 | 1.34 ±2.00 | −0.47 ±1.08 | 0.02 ±0.71 |
| p-value | 0.5781 | 0.4219 | 0.2188 | 0.5 | 0.2813 | 0.5 | 0.2813 | 0.7813 | 0.3422 | 0.0157 | 0.7188 | 0.5 |

Table 5: Error rate on 6 folds of CIFAR-10 with varying number of randomly selected labels of our runs of FixMatch without and with the linear smoothness factor. *Our runs.

We assert that a good response to labeling images should be a built-in feature of SSL algorithms. More precisely, consider two splits

$$\mathcal{L} \cup \mathcal{U} = \mathcal{D} \text{ and } \mathcal{L}' \cup \mathcal{U}' = \mathcal{D} \tag{24}$$

of the same dataset $\mathcal{D}$ with $\mathcal{L} \subsetneq \mathcal{L}'$ (and consequently $\mathcal{U}' \subsetneq \mathcal{U}$), cf. §3.1 for notation. All other parameters being equal, the performance on the test-set of a model trained on the split $\mathcal{L}' \cup \mathcal{U}'$ should be expected to

Effect of adding 10 labels in
CIFAR-10 with random sampling

| labels | 40 ← 50 | 50 ← 60 | 60 ← 70 | 70 ← 80 | 80 ← 90 | 90 ← 100 | 100 ← 110 | 110 ← 120 | 120 ← 130 | 130 ← 140 | 140 ← 150 |
|---|---|---|---|---|---|---|---|---|---|---|---|
| FixMatch* | 1.54 ±3.37 | 3.29 ±9.45 | 3.52 ±3.66 | −2.12 ±4.41 | 0.88 ±1.28 | −0.30 ±1.78 | 2.07 ±2.93 | 1.48 ±3.18 | −0.67 ±2.41 | 1.04 ±2.50 | 0.02 ±0.85 |
| p-value | 0.15625 | 0.65625 | 0.03125 | 0.71875 | 0.15625 | 0.57813 | 0.03125 | 0.5 | 0.42188 | 0.28125 | 0.5 |
| Smooth FixMatch | 2.14 ±10.27 | 6.29 ±9.83 | 1.42 ±5.49 | −0.41 ±1.42 | −0.83 ±3.27 | 0.51 ±1.50 | 0.06 ±4.05 | 2.39 ±3.59 | 0.61 ±0.82 | −0.77 ±0.96 | 0.50 ±0.92 |
| p-value | 0.34375 | 0.42188 | 0.21875 | 0.78125 | 0.34375 | 0.25009 | 0.28125 | 0.34375 | 0.07813 | 0.93099 | 0.34375 |

Table 6: Mean gain and p-value for the improvement on the error rate on 6 folds of CIFAR-10 with 10 labels added incrementally. *Our runs.

be worse than the performance of the same model trained on $\mathcal{L} \cup \mathcal{U}$ only with a small probability. This is a notion of "probabilistic monotonicity" of the training strategy with respect to the labeled dataset.

We were unable to find even a heuristic as to why Consistency Regularization and Pseudo-Labeling should satisfy such a natural property. This does is not directly related to the discontinuity of the loss function, even if the smooth version seems to respond better when the labeled dataset is expanded.

Table 6 shows that the smooth version of the model is more likely to respond correctly to the addition of labeled examples. The p-value for decreasing the error rate is above 0.5 only once for the smooth version, at the passage $70 \rightarrow 80$ labels. Its is above 0.5 for the non-smooth version 3 times, the last one being observed at the passage $90 \rightarrow 100$ labels, and remains high even at the passage $120 \rightarrow 130$ labels, while the one of Smooth FixMatch is already $< 0.08$ but deteriorates again at $130 \rightarrow 140$, even though it remains $< 10\%$.

We can remark, however, that both methods, already with 60 labels have enough information to classify more or less correctly 9 out of 10 classes of CIFAR-10. The difference between a model converging to an accuracy $\geq 92\%$ and one converging to an accuracy $\leq 88\%$ can be read in the confusion matrix: low accuracy is obtained when one class is totally collapsed, while high accuracy when all diagonal entries are significantly $> 0$.

Already in the 80-label benchmark, Smooth FixMatch displays two folds that manage to populate all 10 diagonal elements of the confusion matrix, while FixMatch populates only 9/10, leading to an accuracy smaller than 90%.

## 5    Conclusions

We imposed smoothness of the loss function in PL as it is applied in CR. Our improvement is generic, entails no additional hyperparameters, marginal computational cost and uses no assumption on label distribution, making it more robust than the SOTA when mild misalignment between label distribution between annotated and non-annotated data is introduced. When applied on FixMatch, our method significantly improves the performance in the regime where very weak supervision can achieve performance closer to the fully supervised baseline.

The factor imposing smoothness on the loss function can be interpreted as a measure of the confidence of the network producing the pseudo-labels on the pseudo-labels, ranging from 0 to 1 in a continuous fashion.

Our improvement becomes significant in the limit of weak supervision and performance close to the fully supervised baseline. In the present state of the art, it is therefore significant only in simple datasets, since for datasets close to real-life applications our techniques either demand strong supervision (which overpowers the instabilities caused by the discontinuities in the derivative of the loss), or their performance stays far from the fully supervised baseline, where the instabilities do not manifest themselves and volatility of the performance remains low.

The issue of instabilities should not be expected to be fixed miraculously when moving on to more difficult datasets. If anything, instabilities should be expected to be more important when the limit of weak supervision and high performance is reached in more difficult settings. This work is therefore a word of caution for the direction that SSL should take, in order to anticipate this kind of difficulty when the limit is reached.

Replacement of a step-wise function by a ReLU factor wherever thresholding is involved could lead to improving performance in other contexts.

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

# A  Detailed results of experiments

We provide the full tables of all experiments reported in the paper. The title of each paragraph is the same as the one in the paper with reference to the corresponding table.

In tables corresponding to ablation studies, default parameters are in bold.

All experiments were run using the codebase of Zhang et al. (2021).

## A.1  CIFAR-10-40

Detailed results reported in table 7.

| Fold | FixMatch* | FlexMatch* | Smooth FixMatch | Smooth FixMatch-sq[†] | Smooth FixMatch-sqrt[††] |
|---|---|---|---|---|---|
| 0 | 9.77 | 5.33 | 6.25 | 7.14 | 6.27 |
| 1 | 7.43 | 5.13 | 7.07 | 5.36 | 6.39 |
| 2 | 7.48 | 5.13 | 5.45 | 5.22 | 7.14 |
| 3 | 7.36 | 14.73 | 6.32 | 7.48 | 6.48 |
| 4 | 15.60 | 5.07 | 11.44 | 14.08 | 13.51 |
| 5 | 8.01 | 5.32 | 6.47 | 6.96 | 6.54 |
| | 9.28 | 6.79 | 7.24 | 7.71 | 7.72 |
| | ±2.95 | ±3.55 | ±1.94 | ±2.98 | ±2.60 |
| max | 15.60 | 14.73 | 11.44 | 14.08 | 13.51 |
| min | 7.36 | 5.07 | 5.45 | 5.22 | 6.27 |
| range | 8.24 | 9.66 | 5.99 | 8.86 | 7.24 |
| gain wrt FixMatch | | 2.10 | 2.04 | 1.57 | 1.55 |
| | | ±5.68 | ±1.27 | ±0.91 | ±1.02 |
| p-value | | 0.15625 | 0.015625 | 0.03125 | 0.015625 |

Table 7: Error rate on 6 folds (with each fold result) of CIFAR-10 with 40 labels of our runs of FixMatch without and with the smoothness factor. The p-value line corresponds to the result of the Wilcoxon one-sided test for the given method having lower error rate than FixMatch, which is used as baseline (median being significantly positive, lower is better). Range is maximum minus minimum, smaller is better. * Our run using the codebase of Zhang et al. (2021). [†] Factor shape is square, threshold set at 0.82. [††] Factor shape is square root, threshold set at 0.98.

## A.2  Strong supervision regime

Detailed results on CIFAR-100-2500 reported in table 8. The results confirm that, outside the regime of weak supervision coupled with high performance, the smoothing factor has no significant influence on the performance of the model.

For experiments on Imagenet10%, cf. §B.2.

| | C100-2500 | | | | | |
|---|---|---|---|---|---|---|
| Fold | FixMatch*† | FlexMatch*† | Smooth FixMatch† | FixMatch*†† | FlexMatch*†† | Smooth FixMatch†† |
| 0 | 30.27 | 30.48 | 30.04 | 28.84 | 28.65 | 28.84 |
| 1 | 30.18 | 29.84 | 29.57 | 29.29 | 27.65 | 29.50 |
| 2 | 30.61 | 30.19 | 30.73 | 29.30 | 28.51 | 29.30 |
| | 30.35 | 30.17 | 30.11 | 29.14 | 28.27 | 29.21 |
| | ±0.19 | ±0.26 | ±0.48 | ±0.21 | ±0.44 | ±0.28 |
| gain wrt FixMatch | | 0.18 ±0.28 | 0.24 ±0.30 | | 0.87 ±0.59 | −0.07 ±0.10 |
| p-value | | 0.25 | 0.25 | | 0.125 | 0.8413‡ |

Table 8: Accuracy on 3 folds reporting error rate of last checkpoint. * Our runs. † Weight decay rate 0.0005 as for CIFAR-10. †† Weight decay rate 0.001 as in experiments carried out in Sohn et al. (2020); Zhang et al. (2021). ‡ Two out of three are ties, p-value not reliable.

## A.3  Ablation studies

### A.3.1  Shape of continuity factor

Figure 6 compares the linear with the quadratic factor. Detailed results are reported in table 7. As mentioned in the main part of the paper (§4.3.1 and figure 2), Smooth FixMatch outperforms Fixmatch more significantly than FlexMatch. In other words, while FlexMatch may obtain better performances on average (in particular thanks to the *a priori* information on class distribution that is not used in other methods) it is much worse than FixMatch on some runs, while our methods systematically performs better.

### A.3.2  Threshold

Detailed results reported in table 9. Indicative plot of linear and quadratic factors for $\tau = 0.82, 0.91, 0.95$ in fig. 6. Smooth FixMatch with linear shape factor can be seen to have a milder dependence on the value of the threshold.

| $\tau$ | 0.75 | 0.82 | 0.86 | 0.91 | **0.95** | 0.98 | 0.999 |
|---|---|---|---|---|---|---|---|
| FixMatch | | 15.85 | 8.06 | 7.41 | 7.43 | 7.21 | 12.97 |
| Smooth FixMatch | | 15.68 | 5.20 | 7.38 | 7.07 | 6.56 | 8.59 |
| Smooth FixMatch-sq | 7.38 | 5.36 | 7.52 | 7.34 | 5.41 | 7.27 | 14.21 |
| Smooth FixMatch-sqrt | | 8.09 | 7.44 | 6.72 | 6.55 | 6.39 | 8.24 |

Table 9: Ablation study on upper threshold: error rate on the first fold of CIFAR-10-40 when the upper threshold varies.

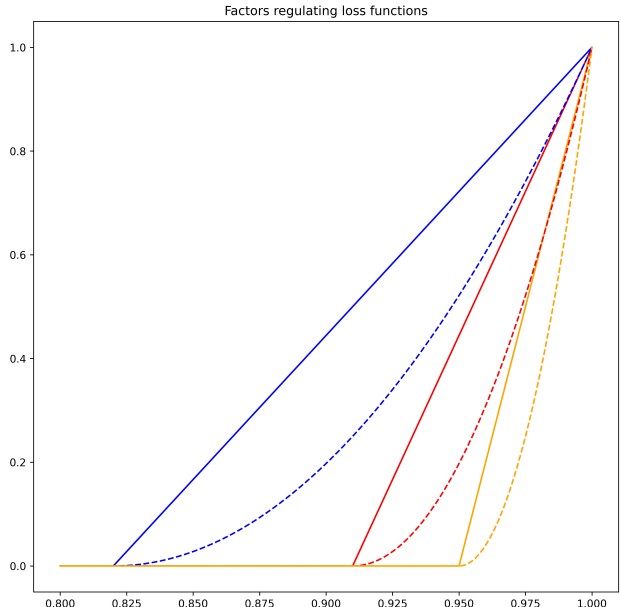

Figure 6: Plot of linear (contiguous) and quadratic (dashed line) factors for $\tau = 0.82, 0.91, 0.95$.

### A.3.3 Class imbalance

We introduce a class imbalance by reducing one class, chosen randomly, to 60%. Detailed results are reported in table 3. When the implicit hypothesis of balanced classes does not hold (even lightly in this case), it is clear that the performance of FlexMatch drops significantly, even below that of FixMatch. On the contrary, our method still exhibits better performance than FixMatch.

### A.3.4 SGD momentum parameter

Detailed results reported in table 11. The default parameter for the rest of our experiments is $\beta = 0.90$.

The value of the momentum has a significant influence on the results. However, the method we propose is less sensitive to this hyperparameter than the original FixMatch.

### A.3.5 Ablation study on initial condition

Detailed results are presented in table 12. Results not included in the aggregate result comparison of §4.4 of the paper.

### A.3.6 Aggregate comparison

In the corresponding section of the paper, the results on FixMatch and Smooth FixMatch contained in tables 7, 9, 3, 11 and 4 are compared. Smooth FixMatch is found to outperform FixMatch at a high significance level, in almost all experiments.

### A.3.7 Factor as loss

We study the effect of learning the smoothing factor by varying the corresponding weight $\lambda_\Phi$ and report results in table 13.

We added the smoothing factor in the total loss with factors 0.0005, 0.005, 0.01, 0.05. This addition amounts to learning the pseudo-label on the weak augmentation, which, as we argue in §C, should contribute to back-propagation with a smaller factor than the unsupervised loss on the strong augmentation.

| Fold | FixMatch | FlexMatch | Smooth FixMatch |
|---|---|---|---|
| 0 | 7.18 | 7.96 | 6.70 |
| 1 | 5.36 | 7.95 | 6.66 |
| 2 | 10.26 | 8.83 | 5.25 |
| 3 | 8.06 | 10.65 | 6.99 |
| 4 | 16.21 | 24.61 | 13.65 |
| 5 | 8.75 | 9.52 | 7.50 |
| | 9.30 | 11.59 | 7.78 |
| | ±3.43 | ±5.90 | ±2.71 |
| max | 16.21 | 24.61 | 13.65 |
| min | 5.36 | 7.95 | 5.25 |
| range | 10.85 | 16.66 | 8.40 |
| gain wrt FixMatch | | −2.28 ±3.05 | 1.51 ±1.94 |
| p-value | | 0.953125 | 0.109375 |
| drop due to imbalance | 0.03 ±1.83 | 4.80 ±7.15 | 0.55 ±0.89 |
| p-value | 0.421875 | 0.109375 | 0.15625 |

Table 10: Ablation study on class imbalance: error rates with one class reduced to 60%. Drop due to imbalance is the difference with respect to the performance of the same model without class imbalance (smaller is better).

| $\beta$ | 0.50 | 0.75 | 0.85 | **0.90** | 0.95 | 0.99 |
|---|---|---|---|---|---|---|
| FixMatch | 33.24 | 33.05 | 30.01 | 15.60 | 14.83 | 34.05 |
| Smooth FixMatch | 30.09 | 29.33 | 15.31 | 11.44 | 13.43 | 34.80 |

Table 11: Ablation study on momentum: error rate when the momentum of SGD varies, reporting error rate of last checkpoint on the fourth fold of CIFAR-10-40.

| seed | 1917 | 2001 | 2019 | **2046** | 2067 | |
|---|---|---|---|---|---|---|
| FixMatch | 17.71 | 22.63 | 13.26 | 15.60 | 15.56 | $16.95 \pm 3.27$ |
| Smooth FixMatch | 13.14 | 13.28 | 13.55 | 11.44 | 13.43 | $12.97 \pm 0.78$ |

Table 12: Ablation study on random seed: error rate on the 4th fold of CIFAR-10-40 when the random seed varies. The default random seed for our experiments is 2046.

| $\lambda_\Phi$ | **0.** | 0.0005 | 0.005 | 0.01 | 0.05 |
|---|---|---|---|---|---|
| Smooth FixMatch | 11.44 | 13.42 | 13.55 | 13.25 | 12.97 |

Table 13: Ablation study on weight of factor: error rate when the weight of factor varies, reporting error rate of last checkpoint on the fourth fold of CIFAR-10-40.

The upper limit $\bar{\lambda}_\Phi$, determined experimentally by calculating the equilibrium values of the relevant losses, was found to be 0.05 for CIFAR-10-40. In a somewhat counter-intuitive way, we found a weight factor of even 1% of this upper limit increased the error rate, and that on the 4th fold of CIFAR-10-40 the error rate was more or less constant for all tested values of $\lambda_\Phi$ (even though it remained below the error rate of FixMatch).

Experiments on other folds with $\lambda_\Phi = \bar{\lambda}_\Phi = 0.05$ confirmed that this value is indeed a tipping point for the error rate, as the experiment on the 1st fold gave an error rate $\sim 14\%$, with FixMatch having an error rate of $\sim 7.5\%$.

The value of $\lambda_\Phi$ was thus set to 0 for all other experiments.

### A.4 Random choice of labeled dataset

We compare Smooth FixMatch with FixMatch on CIFAR-10 with an increasing number of randomly selected labeled images until one algorithm reaches stable performance on par with the fully supervised baseline. We start with 40 labels and proceed with increments of 10 labels. The choice for each labeled fold is made independently, depending only on the random condition, and incrementally, i.e. the fold of 40 labels for the seed 0 is contained by construction in the fold of 50 labels for the seed 0. We present detailed results in an Excel sheet available here. This benchmark is therefore a dummy version of an Active Learning scenario with an increment of the size of one label per class, but random selection.

## B  Details on setup of experiments

Our experiments were run using the codebase of Zhang et al. (2021).

### B.1  CIFAR datasets

The values for hyperparameters used in our experiments are provided in table 14.

Experiments on CIFAR-10 were executed on single GPU P5000 16Go. Experiments on CIFAR-100 were executed on 2 GPUs A100 40Go.

### B.2  Experiments on Imagenet

Training with the standard FixMatch protocol on Imagenet gave equally low accuracy for both FixMatch and Smooth FixMatch, see figure 7. The reason for this low performance is that the network struggles throughout training to learn the labeled dataset, as is made clear by figures 8 and 9. Since accuracy on the labeled dataset is not satisfactory for the greatest part of training, the quality of pseudo-labels, especially in the beginning of training, is low, which results in a high degree of confirmation bias and explains the low performance.

|                      | CIFAR-10 | CIFAR-100 |
|----------------------|:--------:|:---------:|
| Optimizer            | SGD      |           |
| learning rate        | 0.03     |           |
| momentum             | 0.9      |           |
| weight decay         | $5e{-}4$ | $1e{-}3$  |
| ema                  | 0.999    |           |
| $\lambda_u$          | 1.1      |           |
| $\lambda_\Phi$       | 0.       |           |
| $\tau$               | 0.95     |           |
| random seed          | 2046     |           |
| Architecture         | WideResNet |         |
| Depth                | 28       |           |
| Width                | 2        | 8         |
| Num. iterations      | $2^{20}$ |           |
| Labeled Batch-size   | 64       |           |
| Unlabeled Batch-size | 448      |           |
| Folds                | $0-5$    | $0-2$     |

Table 14: Hyperparameter values of Smooth FixMatch for CIFAR datasets. We follow the values used in Sohn et al. (2020) except for $\lambda_u$ which is determined by the calculations of §C

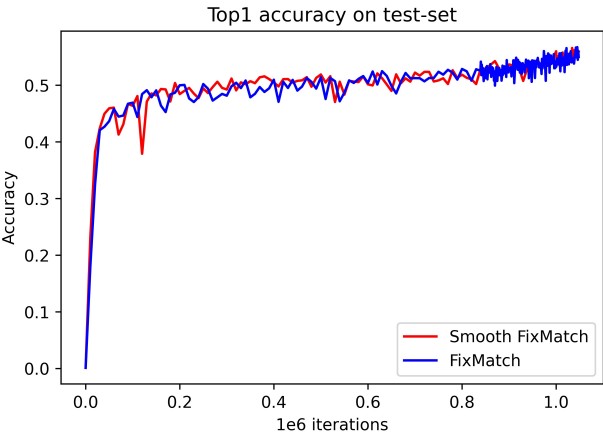

Figure 7: Accuracy on the test-set of Imagenet when training with standard FixMatch protocol.

These results are, for obvious reasons, irrelevant to the (non-)smoothness of the loss function. The algorithm fails to interpolate sufficiently the labeled images, and therefore its extrapolation properties on the unlabeled dataset are irrelevant. The discontinuity in the derivative of the loss function of FixMatch occurred in the unsupervised loss, and our improvement therefore does not improve the interpolation properties of FixMatch. Consequently, our improvement should not be expected to have any impact on performance, which is precisely what we observed in our experiment.

This experiment being of low practical value, we decided against including it in the main part of the article. We only note that in this regime our method does not hinder the performance of FixMatch, as the curves (red for Smooth- and blue for FixMatch) are practically indistinguishable

For completeness, we give a comparative graph of the evolution of accuracy on the labeled part of the batch of the original FixMatch algorithm, trained with standard parameters on CIFAR-10-40, CIFAR-100-2500 and Imagenet-10%. The deterioration is clear and establishes our point.

Experiments on ImageNet were run on one node with 8 GPUs A100 40Go.

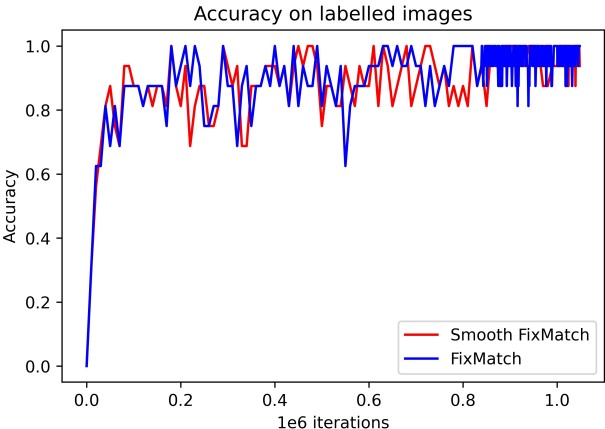

Figure 8: Accuracy on the labelled part of the batch when training with standard FixMatch protocol on Imagenet.

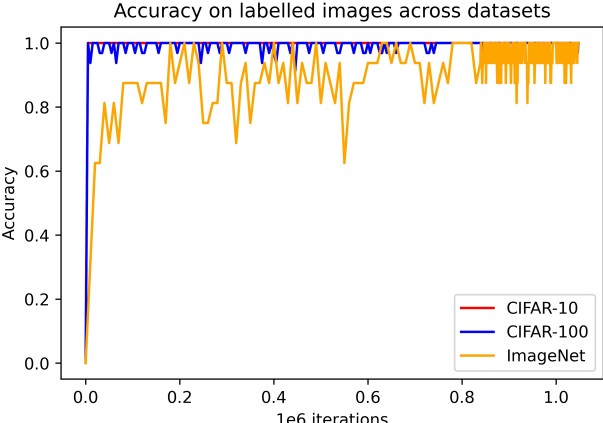

Figure 9: Comparison of the accuracy of FixMatch on the current batch of labelled images across datasets.

## C   Gauging the weight of factor as loss

### C.1   Calculation for $\lambda_u$

We follow notation of §3.8 of the paper and provide the calculation for the factor of the unsupervised loss used in our experiments.

Both the unsupervised losses and $\Phi$ reach equilibrium values very early in training. If $\ell_\Phi$ is the one for $\Phi$, we impose that

$$\lambda_{u,FM}\ell_{FM} = \lambda_{u,SFM}\ell_\Phi\ell_{SFM} \tag{25}$$

The value of $\lambda_{u,SFM}$ for $\lambda_{u,FM} = 1$ can be determined experimentally ($\sim 1.1$ for CIFAR datasets and $\sim 1.8$ for Imagenet) and is therefore not a hyperparameter.

### C.2   Calculation of $\bar{\lambda}_\Phi$

We follow the notation of §3 of the paper.

| | FixMatch | Smooth FixMatch |
|---|---|---|
| Optimizer | | SGD |
| learning rate | | 0.03 |
| momentum | | 0.9 |
| weight decay | | 3e−4 |
| ema | | 0.999 |
| $\lambda_u$ | 1. | 1.8 |
| $\lambda_\Phi$ | | 0. |
| $\tau$ | | 0.70 |
| random seed | | 2046 |
| Architecture | | ResNet50 |
| Num. iterations | | $2^{20}$ |
| Labeled Batch-size | | 128 |
| Unlabeled Batch-size | | 896 |
| Fold | | 0 |

Table 15: Hyperparameter values for Imagenet, for FixMatch and Smooth FixMatch.

Imposing that the derivative for the unsupervised loss be equal to the derivative of $L_\Phi$ gives, in the linear case,

$$\frac{\lambda_{u,SFM}\ell_\Phi}{\ell_{SFM}} = \frac{\lambda_\Phi\ell_\Phi}{1-\tau} \tag{26}$$

The original implementation of FixMatch established that the network benefits more from learning the strong augmentation than the weak one. Therefore, a break in performance should be expected when equality is satisfied, with better performance in the regime

$$\lambda_\Phi < \bar{\lambda}_\Phi = \frac{(1-\tau)\lambda_{u,SFM}}{\ell_{SFM}} \tag{27}$$

The quantity $\bar{\lambda}_\Phi$ is not a hyperparameter, as it can be determined experimentally.

