# OpenReview forum: "Smooth Pseudo-Labeling"
_TMLR — Rejected by TMLR_

### Review · Reviewer_ahYT · 2024-01-15

**Summary Of Contributions:**

This paper studies Semi-Supervised Learning (SSL) with Pseudo-Labels. Based on the finding that the loss function minimized by pseudo-labeling is actually non-continuous and could cause instability of SSL. The authors propose a novel smoothed pseudo-labeling strategy to ensure the derivative of the loss function is monotonic. By applying the proposed method to FixMatch, the authors conducted several quantitative and qualitative experiments to carefully validate the proposed method on robustness, effectiveness, and stability.

**Audience:**

Yes

**Broader Impact Concerns:**

No ethical concerns.

**Claims And Evidence:**

Yes

**Requested Changes:**

Please see weaknesses for details.

**Strengths And Weaknesses:**

Strengths:
- This paper is very intuitive and can provide valuable insight into conducting pseudo-labeling in SSL.
- The experimental result is effective; it constantly outperforms the compared baseline methods.


Weaknesses:
- How the proposed smooth function in Eq. 12 is designed is still intuitive, there is no solid theoretical justification, which is the main concern of this paper.
- The writing could still be further polished. Some demonstrations are not quite clear to me, and it takes time to guess the true intention of the authors.
- The experiments only included simple baseline methods such as FixMatch which is the backbone. How the performance of SPL on other baseline methods is still unclear. To testify the universality of SPL, it is suggested to apply the framework to other SSL methods such as VAT, MeanTeacher, etc. Moreover, it would be better if large-scale datasets such as ImageNet were considered.
- Does such a smoothness factor affect the performance of smoothed pseudo-labelling? Is there a sensitivity analysis to justify the choice of different smoothness factors?
- Missing several closely related SSL literatures:
  - Huang et al., Universal Semi-Supervised Learning
  - Huang et al., FlatMatch: Bridging Labeled Data and Unlabeled Data with Cross-Sharpness for Semi-Supervised Learning
  - Chen et al., Debiased Self-Training for Semi-Supervised Learning
  - Zhang et al., Semi-Supervised Domain Generalization with Known and Unknown Classes
  - Dong et al., Cluster-aware Semi-supervised Learning: Relational Knowledge Distillation Provably Learns Clustering
- Minor problems. By minimizing the margins between columns, Eq. 3 and Eq. 4 could be arranged vertically for reading. The Equation reference is not correct, there are typos such as Eq. Equation XX.

---

### Review · Reviewer_Wsz7 · 2024-02-16

**Summary Of Contributions:**

This paper proposes a semi-supervised learning (SSL) technique named smooth pseudo-labeling. The main motivation is that the original pseudo-labeling (PL) strategy suffers from the accumulated errors from the previous parameters, as argued in Section 3. To solve the problem, this paper introduces a PL strategy using the stop-gradient operation. The main experiment is only conducted on CIFAR-10-40.

**Audience:**

No

**Broader Impact Concerns:**

There is no specific ethical concern.

**Claims And Evidence:**

No

**Requested Changes:**

First of all, the overall paper should be re-written for a better clarity, not only fixing trivial grammar and syntatic errors but also improving the overall writing quality.

Section 3 also should be re-written with more rigouros proofs. For example, it should start from the definition of C0\C1 if this paper keep arguing that it is problematic that the loss function is C0\C1 function. Then the paper should accurately prove the claim (C0\C1 is problem, but C1 is not). The following section should prove that the PL loss function is a C0\C1 function (more rigourosly), and SPL is C1. It is also important that stop gradient is not a proper mathematical function with accurate derivative. It just changes the derivative with different values without assessing any mathematical proof.

Section 4 also needs to be more accurate and rigouros. Many scholars and groups already proved that the existing works are sufficiently reproducible. There already exists open frameworks with reproducibility. The numbers should be conducted on the reproducible framework. Moreover, this paper should contain more comparison methods, especially FlexMatch and SoftMatch. I don't think that the p-value analysis is mandatory for this field, because each experiment usually has very small variance (e.g., less than 0.1 as SoftMatch shows, but the gaps between methods are usually larger than variance).

Overall, this paper should need a heavy revision, including the overall presentation, the theoretical part and the empirical part. I don't think these changes can be happened during the short revision period, and even though it is revised well, I think the revised paper is completely different paper from the original paper, hence it will need another review round.

### Minor

- Throughout the paper, C0 and C1 functions are used without any introduction. Please specify their definition, or avoid to use them without any definition.
- $\theta$ and $\Theta$ are mixed in the paper (e.g., eq 5 and 7 use $f_\Theta$, eq 6 use $f_\theta$). Please unify them.

**Strengths And Weaknesses:**

## Weakness

### Poor writing and presentation

I did my best to understand the manuscript to fairly evaluate the claim and corresponding evidence by the paper. However, it was really difficult to understand the manuscript. This paper has very poor writing and presentation. In my opinion, due to the poor writing quality, no individual in TMLR's audience will be interested in this paper. It does not simply mean that some trivial grammatical errors (e.g., "general and complex notations and notation" in the first paragraph or "PL is C0\C1 has not been" ... in page 2. There are a lot of trivial grammatical and syntactic errors) or poor image and table qualities (Fig1,2,3,4,5 and Tab1,2,3,4), but I mean that the overall presentation is also poor and needs heavy revision.

I personally suspect that this paper was originally written in a 2-columned format, and I just changed the main package to TMLR without any revision. Specifically, Tables 3 and 4 don't have to be rotated, and all figures are too large for the one-column format of TMLR. All citations are not correctly declared by `citep` and `citet`, but look all of them are `cite`. Similarly, I also found `eq. equation 9`, (below eq 9) perhaps it is due to the `eq \cref{reference}`. I don't think the current paper quality is not suitable to be published, and therefore, it is not acceptable to TMLR as well. Also, I don't think this paper needs to have 20 pages long.

However, even though the paper will be revised during the discussion period, I still don't think this paper is acceptable due to the following reasons.

### Insufficient theoretical evidence for "the dependency of $L_{PL}$ on $\theta$"

As far as the reviewer understood, this paper keep arguing that the PL loss function is affected to the previous step parameters, and it makes an accumulated error during the iterative optimization. However, I don't think the mathematics in Section 3.3 is fully convincing.

First, there is no derivation for the equation (10). Eq (9) is trivial from Eq (8) by substituting $i-1$ to $i$ (but, it is actually not a rigorous, because this substitution did not consider the initialization, such as when $i = 0$). However, we cannot generally argue that Eq (10) holds without a rigorous proof. Furthermore, Eq (10) is actually not specifically true for PL loss function, because all $\theta_i$ is indeed the summation of the $\theta_j$ where $j < i$. It means that, the accumulation is not the special problem of PL, but it is the problem of general iterative gradient descent-based algorithms.

Second, the paper keep arguing that it is because of "PL loss is a C0\C1 function". I cannot get any idea why C0\C1 loss function specifically makes the problem. It needs more rigorous justification.

### Insufficient theoretical evidence for "smooth pseudo-labeling has continuous derivative"

The smooth PL is defined by ReLU, which is the exactly C0\C1 function. Therefore it will suffer from the same problem with PL loss. Note that the "stop gradient operator" cannot be argued as a rigorous mathematical differentiable function, because it did just change the definition of the differential derivative without satisfying the mathematical definition of derivative. Hence, the proposed loss function cannot be argued as C0\C1 function.

Also, SPL is also dependent to the $f_\theta$ value. It means that SPL also has the same dependency as PL has.

Overall, this paper has critical flaws in the theoretical evidence:

1. There is no proof for C0\C1 function leads to an error accumulation, but C1 function does not
2. There is no rigorous proof for SPL function is C1 function. As I mentioned, stop gradient cannot be defined as Cn functions because it twisted the definition of the derivative.

It means that this paper is not accurately supported and therefore it does not satisfy the TMLR evaluation criteria.

### Insufficient empirical evidence.

First of all, I don't think only showing CIFAR-10-40 is sufficient empirical evidence. As many other SSL papers do, I think this paper needs CIFAR-10-40, -250, and -4000, CIFAR-100-400, -2500, and -10000, SVHN-40 and -1000, and STL-10-40 and -1000.

Moreover, there are not enough comparison methods to support the effectiveness of the proposed method. In my opinion, this paper should contain more comparison methods, such as PL, MeanTeacher, MixMatch, ReMixMatch, UDA, FixMatch, Influence, FlexMatch and SoftMatch. I didn't follow up more recent methods, but I certainly believe that there will be more recent works since SoftMatch (ICLR'2023 SoftMatch: Addressing the Quantity-Quality Tradeoff in Semi-supervised Learning).

Finally, the numbers are much worse than the other papers. For example, FixMatch CIFAR-10-40 shows 7.47$\pm 0.28$ error in the other papers (see SoftMatch Table 2), but this paper shows 9.28$\pm 2.95$. It means that the evaluation by this paper is not convincing enough. I cannot agree with the claim:

> "We note that, while the results for FixMatch and Smooth FixMatch were reproducible after some necessary modifications assuring control of batch composition by the random condition, we were unable to get reproducible results for FlexMatch."

This field is widely studied for many scholars and many groups. There are many reproducible frameworks for each method. For example "USB: A Unified Semi-supervised learning Benchmark" (https://github.com/microsoft/Semi-supervised-learning).

---

### Review · Reviewer_L4kT · 2024-03-02

**Summary Of Contributions:**

To alleviate the issue of discontinuities in derivatives of the pseudo-labeling loss function, this paper introduces a Smooth Pseudo-Labeling (SPL) loss function for SSL. It borrows ideas from a ReLU function to have a continuous derivative and improves over FixMatch.

**Audience:**

Yes

**Broader Impact Concerns:**

There are no obvious ethical issues.

**Claims And Evidence:**

No

**Requested Changes:**

See 12 weaknesses above.

**Strengths And Weaknesses:**

**Strengths:**

1. Describing SSL as a signal-to-noise ratio problem is interesting, where the signal comes from labeled data and the noise is from unlabeled data.

2. The analysis and discussion on particular methods are elaborate, e.g., FixMatch, FlexMatch, etc.

3. The smoothing is crucial for improved pseudo-labeling/self-training, which is the quest of this paper.

**Weaknesses:**
1. The paper writing should be improved to make the structure clearer and the logic more rigorous.

2. It would be better to show the instabilities (e.g., standard deviation) in performance when labels are very scarce, e.g., 4 labels per class on CIFAR-10, by varying random seeds, hyperparameters, or training parameters. Moreover, a comprehensive comparison of performance instability for the proposed method and existing ones is encouraged.

3. As the number of labels increases, will the drawback disappear that the pseudo-labeling loss function has discontinuities in its derivatives? Are the discontinuities caused by a class imbalance in sample selection that skips significant landscapes or fragile features and decision boundaries due to a lack of supervision for learning intended patterns? An in-depth analysis is encouraged.

4. The abstract should be self-contained. The authors should provide a high-level summary of technical details that may be one or two sentences, to demonstrate that the proposed method is technically sound, e.g., how to achieve smooth pseudo-labeling (SPL).

5. It is certainly good to use the representative FixMatch as the baseline method for testing the proposed improvements. However, there are several state-of-the-art (SOTA) methods in SSL literature that rely on self-training with pseudo labels, like STOCO [a], FullFlex [b], etc. It is encouraged to test the proposed improvements on them as well for new SOTA performance.

[a] Tang et al. Stochastic Consensus: Enhancing Semi-Supervised Learning with Consistency of Stochastic Classifiers. ECCV, 2022.

[b] Chen et al. Boosting Semi-Supervised Learning by Exploiting All Unlabeled Data. CVPR, 2023.

6. The introduction part should be refined. The authors should provide a brief analysis of closely related works, accurately summarize the drawbacks of them, and clearly motivate the present idea and technique.

The authors should provide a more compact, refined, and complete review of related works. The proposed method falls into the realm of traditional SSL, rather than unsupervised pretraining based SSL. For the traditional SSL, the authors should summarize several mainstream technical paths like in [c] and may discuss methods along the line of FixMatch in more detail.

[c] Tang and Jia. Towards Discovering the Effectiveness of Moderately Confident Samples for Semi-Supervised Learning. CVPR, 2022.

7. To what extent do existing methods for the selection of pseudo labels reduce the noise as the training proceeds? The noise rate analysis in [a] may be needed.

8. Can we see the discontinuity of the derivative of the pseudo-labeling loss function in a 3D space through some visualization tools and techniques?

9. CIFAR-100 has the same dataset size as CIFAR-10, i.e., 60K, but more classes, i.e., 100, forming a much more difficult benchmark. On this basis, the phenomenon of performance instabilities should be more obvious. However, the authors observe a reverse phenomenon. Such a result should not just be attributed to the performance gap between an SSL method and a fully-supervised upper bound. More analysis is required, e.g., each instance in each class of CIFAR-100 is semantically representative.

10. In Eq. (13), the SPL function acts as a weighting scheme, with the weight given by Eq. (12), i.e., zero or a value directly proportional to predicted confidence. Till now, there have been many works using confidence, entropy, or uncertainty based weighting schemes in SSL or unsupervised domain adaptation (UDA). The authors should discuss and compare those methods with SPL.

11. The results on CIFAR-100 are important and thus the authors should move them to the main text.

12. In experiments, the authors should involve in comparison more representative and advanced methods of various technical lines. The tables should be placed horizontally for readability and the figures should be at an appropriate scale to avoid image blur.

---

> ### Author Response · Authors · 2024-03-13
>
> We wish to thank reviewer L4kT for their much appreciated feedback. We will do our best to respond to their critics in the upcoming revision of the paper. The remarks not explicitly mentioned in our response will be taken into account in our revision as presented by the reviewer and we omit reference to them in our response just for the sake of brevity.
>
> Concerning the weaknesses remarked by the reviewer:
> 1. We are revizing the paper to improve clarity, but we would appreciate if the reviewer could point to specific problems in logic and rigor
>
> 2. An ablation study on the random seed is presented in section A.3.7 of the appendix, with 4 additional r.s. Ablation studies also include the momentum (4.3.4) and the PL threshold (4.3.2). Results can be moved from the appendix to the main text if the reviewer thinks it relevant.
>
> 3. We performed an extensive study of the influence of adding 10 labels in 4.5, even though not in the apriori class balanced labeled dataset. It shows that the behavior of both algorithms is not monotonic, but SPL behaves slightly better. The discontinuities are not caused by class imbalance, as they are present already in that scenario. They are a built-in caracteristic of any loss function using brutal thresholding, such as PL and all methods constructed upon it.
>
> 5. STOCO does not claim a SOTA on CIFAR-10-40, and their results are indeed on a par with ours but using considerably more resources. Moreover, the method presents a strange behavior of the algorithm with respect to the number of stochastic classifiers, cf. table 1 of the paper, and no sensitivity studies with respect to hyperparameters are presented. We could not find an explicit mention of the number of runs, but it appears to be 5. However, their results on FlexMatch, 4.99±0.16, disagree with ours, 6.79 ±3.55.
>
> Boosting SSL, even though an interesting method, runs experiments only on 3 folds, a protocol that we found to be problematic in our study of the scarce-label regime, e.g. they give better results for FixMatch than the original paper, or ours for that matter.
>
> Unfortunately we do not have the time to run a comparison.
>
> 8. Our attempts produced a messy result, precisely due to the discontinuity...
>
> 9. We are not certain what the reviewer means by "reverse". The two methods appear to be equivalent, precisely due to the low volatility of the baseline in the benchmark. The benefits from smoothing the loss function are expected to be important when volatility is high, as we tried to explain in the paper. A quick class-wise analysis showed that many classes are collapsed, which cannot be remedied without passing on to a more complex method (or starting with a pretrained backbone). We could not understand the remark "More analysis is required, e.g., each instance in each class of CIFAR-100 is semantically representative". If we understood correctly, the collapsing of classes, even with 25 labels per class, points to the opposite direction.
>
> 10. SPL can be interpreted as a use of confidence in weighing, indeed. However, methods such as ConMatch are in the direction of adding modules and thresholding, which makes smoothing out discontinuities more delicate. It is also in the direction of adding complexity to the baseline, which we tried to avoid. Added a brief discussion in the related work section.
>
> 12. The tables could not be made to fit horizontally while keeping readability or without splitting in two parts, which we thought undesirable. They hardly fit the page vertically...

---

### Review · Reviewer_CRd5 · 2024-03-03

**Summary Of Contributions:**

The paper addresses issues in Semi-Supervised Learning (SSL), particularly regarding Pseudo-Labeling (PL) methods. It introduces Smooth Pseudo-Labeling (SPL) to mitigate instabilities caused by discontinuities in the loss function derivatives. Through experiments on FixMatch, the study demonstrates significant performance improvements, especially in scenarios with limited labeled data, without additional complexity. Additionally, a new benchmark is introduced, highlighting that SSL algorithms do not necessarily see performance improvements when labeled images are added.

**Audience:**

Yes

**Broader Impact Concerns:**

I have no concerns about the broader impacts and the ethical implications.

**Claims And Evidence:**

Yes

**Requested Changes:**

Please refer to the Weaknesses above.

**Strengths And Weaknesses:**

Strengths:
1. The paper is well-organized. To my knowledge, the research problem is interesting.
2. The motivation for introducing smooth pseudo-labeling is natural and acceptable.
3. The writing of this paper is generally good.

Weaknesses:
1. Section 1 is a bit long. New readers may feel confused in understanding the main contributions of this paper. The authors may consider 1) trimming Section 1; 2) summarizing their contributions into several points; and 3) comparing 2) with existing works in a conspicuous place.
2. The resolution of Figures 2, 3, 4, 5 is low.
3. The authors are encouraged to evaluate their method in more datasets in addition to CIFAR.
4. In addition to semi-supervised learning, self-supervised learning (+ fine-tuning) is also an effective technique to address insufficient annotations. I think it may be valuable to present some discussions on this issue, aiming to highlight the necessity of this paper.

---

> ### Author Response · Authors · 2024-03-13
>
> We wish to thank reviewer for their much appreciated feedback. We will do our best to respond to their critics in the upcoming revision of the paper. All remarks except remark num. 3 will be taken into account in our revision as presented by the reviewer.
>
> Concerning the comment num. 3 on the fact that we evaluated our algorithm mainly on CIFAR-10, we invite the reviewer to consult the response to all reviewers, published under the rolling discussion.

---

### Comment · Action_Editor_UxUW · 2024-03-03
**Rolling discussion**

Dear authors,

We have collected reviews for the submission. There are two weeks for rebuttal before the reviewers submit their final recommendation.

Best wishes,
AE

---

> ### Author Response · Authors · 2024-03-13
>
> We would like to thank all reviewers for the time and effort spent reading our paper and coming up with helpful remarks.
>
> We would like to address here the recurrent remark concerning the lack of comparison with methods other than FixMatch and the focus on CIFAR-10.
>
> Concerning the latter issue, while preparing a paper, one can perform only so many experiments. We made a conscious decision to restrict the scope and focus on depth, by investigating one a single dataset. We thus performed a considerable number of experiments on CIFAR-10 with 40 labels, including quite extensive ablation and sensitivity studies that seem to be absent from a large number of papers, including some of those suggested by the reviewers. In a large part of the literature, more methods are indeed compared under the same conditions, but
>  on only 3 runs per experiment, something that our more focused and extensive studies show to be insufficient for definite conclusions in the high volatility regime.
>
> We would also like to note that tab. 3 of the paper, and fig. 5 which we consider to be an important finding, demanded 12x6x2 = 144 individual runs (6 folds for two methods tested on 12 benchmarks). Tab. 4 and fig. 5 came as a surprise, as they show that the behavior of Smooth FixMatch and FixMatch are not the ones that anyone, including the authors, would naively expect as they lack monotonicity when labels are added incrementally. This finding, coupled with the unexpected sensitivity of FixMatch with respect to the SGD momentum, see fig. 3 of the paper, would have been out of reach had we decided not to restrict the scope of our experimental investigation. Even though we understand the issue raised by the reviewers, we think that the trade-off for the community in this case is clearly positive.
>
> Unfortunately, we don't have the time to benchmark other methods in the two weeks that we are given for the rebuttal. We would like to remark, however, that, the experimental validation of Boosting Semi-Supervised Learning lacks, in our opinion, some critical ablation studies whose equivalents we did our best to include in our study, since they test only on 3 folds while no ablation on the threshold or the SGD parameters are performed. The STOCO method does run tests on 5 folds, but a short and, in our opinion, not sufficiently thorough analysis on the optimal number of classifiers is presented (the main contribution of the paper being the addition of classifiers), while sensitivity analysis on other factors is totally absent. If we were to include comparison with such methods and test them on par with our own, we would need to add sensitivity analyses and this would have added a considerable amount of experimental work, precluding us from correctly backing our own work with experimental results.
>
> In conclusion, we think that our findings are, in the essence of things, as well backed by experiments as expected in the literature, and that, in some aspects, they are substantially better backed by experiments than the other methods that reviewers proposed to compare with. We also think that taking up the additional work of properly testing these methods and making up for the insufficiencies of the papers would have led us astray from our analysis.

---

### Decision · Action_Editor_UxUW · 2024-05-04

**Recommendation:** Reject

**Comment:**

This paper targets an important problem that how to further improve existing semi-supervised methods. A new pseudo-labeling loss function is proposed, which may encourage smoothness. It adds a multiplicative factor in the loss function that smooths out the discontinuities in the derivative due to thresholding.

However, there are some common concerns about claims and evidence, including concerns about the smooth property of the proposed loss function, missing important baselines, and lacking sufficient datasets. By considering these concerns, a major revision is needed. However, because the open review system does not have the option of major revision, we regret to reject the paper for now and encourage resubmission. The AE believes that this paper would be strong after addressing these concerns.

**Audience:**

How to further improve existing semi-supervised methods can be an interesting problem and has many audiences. However, for the benefits of this paper, it would be great to address the major concerns to further improve the claims before submission.

**Claims And Evidence:**

Some major claims made in the submission are not well supported as mentioned cross different reviewers, which are as follows
1. The justification of the smoothness of the designed function.
- Reviewer Wsz7: “The smooth PL is defined by ReLU, it will suffer from the same problem with PL loss.”
- Reviewer ahYT, the proposed smooth function in Eq. 12 is designed is still intuitive and lack of solid theoretical justification, ”

2. The empirical evidence provided is insufficient, both in terms of dataset coverage and comparison methods.

- Reviewer Wsz7: “this paper should contain more comparison methods, such as PL, MeanTeacher, MixMatch, ReMixMatch, UDA, FixMatch, Influence, FlexMatch and SoftMatch.”
- Reviewer L4kT “I think this paper needs CIFAR-10-40, -250, and -4000, CIFAR-100-400, -2500, and -10000, SVHN-40 and -1000, and STL-10-40 and -1000.”
- Reviewer L4kT: “ there are several state-of-the-art (SOTA) methods in SSL literature that rely on self-training with pseudo labels, like STOCO [a], FullFlex [b], etc. It is encouraged to test the proposed improvements on them as well for new SOTA performance.”

**Resubmission Of Major Revision:**

The authors may consider submitting a major revision at a later time.

---

> ### Author Response · Authors · 2024-05-22
> **Regarding the justification of the smoothness of the designed function**
>
> While in full respect of the editorial authority to determine publication, we feel strongly that there remains a serious matter we need to raise.
>
> If the paper is to be rejected (even if partially so) on the grounds that the proposed loss function is not smooth, we expect the reviewer who raised the concern without backing it with a calculation, or the AE who accepted the validity of the concern at face value to be able to produce a set of parameters in the set of discontinuities of the first derivative of the function and a calculation establishing the discontinuity. We do not understand how there can be "*concerns about the smooth property of the proposed loss function*". A function either is smooth or it is not. This is a matter of fact, well established and, we dare say, trivial in scholarship; not a matter of opinion, ongoing debate, or cutting-edge research. More importantly, we wish to correct the error, if indeed there is one.
>
> In the paper we argued that the function of eq. 14 is smooth and that it is $C^1$ equivalent to the one of eq. 18. We have explained that $\partial_{\theta} \phantom{,} \mathrm{sg}(f_{\theta}(x)) \equiv 0$ and $ \mathrm{sg}(f_{\theta}(x)) \equiv  f_{\theta}(x)$ which is the definition of the stop gradient operator and the way it is implemented in all autograd libraries known to the authors.
>
> We ran the following Pytorch function on a dense grid of parameters and it did verify our calculation. The function of eq. 17 is $C^1$ and this is highschool-level calculus. Its derivative is equal to the derivative of $\mathrm{ReLU} (\frac{\mathrm{sg} (f_{\theta}(x))-\tau}{1- \tau} ) \log ( f_{\theta}(x) )$ (admittedly with a sign error that will be corrected, but this is irrelevant for the study of the continuity of the derivative). This is at most third-year calculus in most engineering schools. The fact that the derivatives are equal was verified by the authors by hand, on WolframAlpha and numerically, and the code is provided at the end of our comment for reproducibility.
>
> The function of eq. 17 is implemented as "closed_form_loss" in the snippet, and the function $\mathrm{ReLU}  (\frac{\mathrm{sg} (f_{\theta}(x))-\tau}{1- \tau} )  \log ( f_{\theta}(x) )$ as "sg_loss". The code is exceedingly verbose in order to cast aside any possible doubt regarding the calculation.
>
> It should be stressed that these calculations add nothing significant to the arguments presented in the paper, and, more importantly, nothing that should have a place in a research-level paper.
>
> We would truly appreciate an answer, either from the AE or from reviewer Wsz7, if the terms of scientific dialogue are to be respected. We expect either a calculation pointing out our error and providing a set of parameters in the set of discontinuities of the proposed loss function, or an acknowledgment of the correctness of our argument.
>
> We thank you in advance for your response.
>
> ```
> import torch
>
> @torch.no_grad
> def no_grad_ReLU(x,t):
>     return torch.maximum(x - t, torch.zeros_like(x)) / (1 - t)
>
>
> def loss_der_dif(s, t):
>     x = s.clone().detach().requires_grad_(True)
>     s.requires_grad_(False)
>     sg_loss = no_grad_ReLU(x,t) * torch.log(x)
>     closed_form_loss = (torch.ge(s, 0) * torch.le(s, t) * (1 - t + t * torch.log(t)) +
>                         torch.le(s, 1) * torch.ge(s, t) * (1 - x + t * torch.log(x))) / (t-1)
>     (closed_form_loss - sg_loss).backward()
>     return x.grad
>
> if __name__ == '__main__':
>     points = [[s,t] for s in range(1,1000) for t in range(1,1000)]
>     for point in points:
>         s = torch.tensor(point[0]/1000)
>         t = torch.tensor(point[1]/1000)
>         grad = torch.abs(loss_der_dif(s, t))
>         if torch.ge(grad,0.0001):
>             print(f"trouble at s={s}, t={t}, grad = {grad}")
>
> ```
>
> At the level of pure technicality, the functions ```torch.ge() & .le()``` in the definition of ```closed_form_loss``` should indeed take a copy of x with requires_grad_ set to False. Replacing the relevant line by
> ```
> closed_form_loss = (torch.ge(x, 0) * torch.le(x, t) * (1 - t + t * torch.log(t)) +
>                     torch.le(x, 1) * torch.ge(x, t) * (1 - x + t * torch.log(x))) / (t-1)
> ```
> produces the same result.
>
>
> The formal calculation can be carried out on WolframAlpha using the following sequence of commands. The free [online version](https://www.wolframalpha.com/) is sufficient.
>
> 1. ```derivative of (y-t) log(x)/(1-t) with respect to x``` where y is used as a placeholder for $\mathrm{sg}(x)$. Result: $(t - y)/((-1 + t) x)$
>
> 2. ```(t - y)/((-1 + t) x) for y = x``` in order to substitute the placeholder for the value before integration. Result: $(t - x)/((-1 + t) x)$
>
> 3. ```integrate (t - x)/((-1 + t) x)``` in order to obtain a primitive that is $C^1$ equivalent to $\mathrm{sg}((x-t)/(1-t))\log(x)$. Result: $(-x + t \log(x))/(-1 + t) + $ constant.
>
> This is the loss function as proposed in the paper (again, modulo the typo replacing $-1+t$ by $t-1$ in the denominator).

---

> > ### Comment · Reviewer_Wsz7 · 2024-05-30
> >
> > Dear authors,
> >
> > First of all, it would be better if the authors could provide the answer during the discussion period, not after the overall process is over. I had no response from the authors, and none of my concerns were addressed in the revised paper.
> >
> > Second, ReLU is a mathematically non-differentiable function. It is 1st-grade university calculus. This is because the derivative is defined by the concept of the limit of function. If you take the limit of ReLU'(0) from the left side, the limit value will be zero; if you take it from the right size, the limit will be one. It means that ReLU has no unique definition of the derivative at "0". Yes, PyTorch and other AutoDiff-based libraries just take "0" for this undefined value, but it does not mean that ReLU is differentiable; we take a proxy for calculating the derivative of ReLU. "ReLU is not differentiable" and "PyTorch computes ReLU backward" are not equivalent arguments.
> >
> > You can exactly see the example even on the Wikipedia page for "smoothness" https://en.wikipedia.org/wiki/Smoothness
> > ReLU is continuous, but not differentiable at x=0, so it is of class C0, but not of class C1.
> >
> > Similarly, stop gradient is another proxy function of computing the derivative, which is not the correct definition of the derivative. Yes, we can update a function with any update rule, but it still does not mean that it is a correct derivate function for the given function.
> >
> > Finally, there were also many other concerns raised by me and the other reviewers. It is not the only reason for the rejection. Please consider revising the paper according to the reviewers' advice.

---

> ### Author Response · Authors · 2024-07-01
>
> We take note of the decision. Although we disagree on a number of points, we find it problematic to consider that "The justification of the smoothness of the designed function" is a valid argument. Apart from the fact that, for example, reviewer ahYT has no complaints about the smoothness itself, the point of reviewer Wsz7 is only based on the presence of ReLU in the proposed loss.
>
> Reviewer Wsz7 merely repeated the trivial fact that ReLU is non-differentiable at 0, which is beside the point since, for example, both
> $$\mathrm{ReLU}(x^3) \text{ and }\mathrm{ReLU}(x)^3$$
> are differentiable (in this case, the same function). The argument that the reviewer has put forward, namely that the mere presence of ReLU mechanically implies non-smoothness, is not a valid mathematical argument, since $\mathrm{ReLU}$ is post- or pre- composed with the $\mathrm{sg}$ operator in the definition of the loss function.
>
> The calculation of the derivative of the loss function is as follows:
> $$
> \begin{array}{r@{}l}
> \frac{\mathrm{d}}{\mathrm{d}x} \left( \mathrm{sg}_x \mathrm{ReLU}(\frac{x-\tau}{1-\tau})) \log(x) \right) &=  \log(x)\frac{\mathrm{d}}{\mathrm{d}x} \mathrm{sg}_x(\mathrm{ReLU}(\frac{x-\tau}{1-\tau}))+
> \mathrm{sg}_x(\mathrm{ReLU}(\frac{x-\tau}{1-\tau}))\frac{\mathrm{d}}{\mathrm{d}x} \log(x)
> \\\\
> &= 0 + \mathrm{sg}_x(\mathrm{ReLU}(\frac{x-\tau}{1-\tau}))\frac{1}{x}
> \\\\
> &=   \mathrm{sg}_x(\mathrm{ReLU}(\frac{x-\tau}{1-\tau}))\frac{1}{x},
> \end{array}
> $$
> a continuous function, where the first term of the development of the product rule vanishes by the very definition of $\mathrm{sg}_x$. This has been repeatedly explained in full rigor both in the article and our comments, and is in total agreement with common knowledge in the community and the implementation of stop-gradient operations in all autograd tools in use known to the authors. It is a special case of the definition of the $\mathrm{sg}$ operator as given [in our response to the review](https://openreview.net/forum?id=49k4PhQQ6E&noteId=CY1ZlUmH8c), which the reviewer will most surely find useful for future reference.
>
> We consider the length of the discussion about such a trivial issue very problematic and will not continue the discussion any further.